

SciPost Phys. Lect. Notes 95 (2025)

# Simulation of the 1d XY model on a quantum computer

Marc Farreras[1,2]⋆ and Alba Cervera-Lierta[2]†

**1** Leiden Institute of Advanced Computer Science (LIACS),
Leiden University, Leiden 2333 CA, The Netherlands
**2** Barcelona Supercomputing Center, Plaça Eusebi Güell, 1-3, 08034 Barcelona, Spain

⋆ m.farreras.i.bartra@liacs.leidenuniv.nl , † alba.cervera@bsc.es

## Abstract

The field of quantum computing has grown fast in recent years, both in theoretical advancements and the practical construction of quantum computers. These computers were initially proposed, among other reasons, to efficiently simulate and comprehend the complexities of quantum physics. In this paper, we present a comprehensive scheme for the exact simulation of the 1-D XY model on a quantum computer. We successfully diagonalize the proposed Hamiltonian, enabling access to the complete energy spectrum. Furthermore, we propose a novel approach to design a quantum circuit to perform exact time evolution. Among all the possibilities this opens, we compute the ground and excited state energies for the symmetric XY model with spin chains of $n = 4$ and $n = 8$ spins. Further, we calculate the expected value of transverse magnetization for the ground state in the transverse Ising model. Both studies allow the observation of a quantum phase transition from an antiferromagnetic to a paramagnetic state. Additionally, we have simulated the time evolution of the state all spins up in the transverse Ising model. The scalability and high performance of our algorithm make it an ideal candidate for benchmarking purposes, while also laying the foundation for simulating other integrable models on quantum computers.

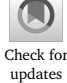

# 1  Introduction

In the first decade of the XXI century, we witnessed an explosion of the quantum computing field driven by the incredible potential that quantum computing exhibits to solve some intractable classical problems [1]. Among these challenges, one of the enduring objectives of quantum computing is the simulation of quantum systems. Although several classical strategies exist for simulating such systems [2,3], they often prove to be inefficient when dealing with complex quantum systems. Consequently, the simulation of quantum systems demands alternative methods for efficient execution. Here, quantum computers emerge as a promising solution, since due to their quantum nature the simulation of strongly correlated systems is the natural arena where quantum computers are expected to show a clear advantage over classical ones, as Feynman stated in Ref. [4].

Despite having undergone considerable development during the last decade, quantum computing is still in an early stage. The current state of quantum computing is known as the Noisy Intermediate-Scale Quantum (NISQ) era [5]. The NISQ era has been characterized by constrained-size quantum processors (containing 100 qubits approximately) with imperfect control over them; they are sensitive to their environment and prone to quantum decoherence and other sources of errors. Despite these challenges, researchers have successfully pushed the boundaries of current quantum technology, particularly in the simulation of physical systems [6]. This progress has been largely enabled by the development of error mitigation techniques and the optimization of quantum circuits [7,8], where more hardware-faithful implementations have been prioritized over error-prone alternatives. Nevertheless, these methods require thorough characterization of the underlying quantum hardware, making it essential to develop scalable and standardized benchmarking techniques. Such benchmarks are crucial for both companies and researchers to evaluate and compare the efficiency of emerging quantum devices.

This paper presents a circuit suitable for the NISQ era, offering the capability to explore intriguing phenomena such as quantum phase transitions. Our work consists of implementing a quantum circuit that performs the exact simulation of a 1-D spin chain with an XY -type interaction. We programmed a set of Python libraries that allows the implementation of the circuit for systems with a power of 2 number of qubits using Qibo [9], an open-source framework for quantum computing. Moreover, Qibo is the native language of the Barcelona Supercomputing Center quantum computer, which will allow the users to directly test this algorithm with real machines. The foundation of our work is based on Ref. [10,11], where the steps followed to design the quantum circuit rest upon tracing and implementing the well-known transformations that solve the model analytically [12]. As a result, this technique can access the whole spectrum, enabling us to simulate any excited or thermal state and its dynamical evolution. In addition, this framework can be easily extended to other integrable models, including the Kitaev-honeycomb model [13], or to systems whose effective low-energy behavior can be suitably described by quasi-particles [14].

This paper is organized as follows: In Sec.2 we describe the characteristics of the XY model and solve it analytically. Moving to Sec.3, we revisit the method introduced in Ref. [10] to construct an efficient circuit that diagonalizes the XY Hamiltonian. We then present the circuit employed for simulating spin chains of $n = 4$ and $n = 8$ qubits. Next, in Sec.4 we design a quantum circuit tailored for exact time evolution. Our simulations, utilizing the proposed quantum circuit, are detailed in Sec.5. Finally, the conclusions are exposed in Sec.6 and the code is available in Ref. [15].

## 2 The $1-D$ XY model

The XY model is derived from the Heisenberg model [16] by introducing an easy-plane anisotropy. Those models are widely used to study critical points and phase transitions of magnetic systems within the condensed matter field. The $1-D$ XY Hamiltonian can be written as

$$\mathcal{H}_{XY} = J\left(\sum_{i=1}^{n} \frac{1+\gamma}{2}\sigma_i^x\sigma_{i+1}^x + \frac{1-\gamma}{2}\sigma_i^y\sigma_{i+1}^y\right) + \lambda\sum_{i=1}^{n}\sigma_i^z, \tag{1}$$

where $n$ is the number of spins in the 1-D spin chain, $\sigma_j^i$ with $i = x, y, z$ are the Pauli matrix acting on the site $j$, $J$ determine the behavior of the ordered phase, ferromagnetic for $J < 0$ and antiferromagnetic $J > 0$, $\gamma$ is the anisotropic parameter and $\lambda$ represents the strength of the transverse magnetic field.

One important feature for which the XY model stands out is that it exhibits a quantum phase transition [17, 18]. These transitions occur at zero temperature and stem from the competition of the different terms within the Hamiltonian, regulated by a non-thermal physical parameter of the system. At zero temperature, each term presents a specific ground state, and the properties of these ground states dictate the phase of the system.

Specifically for the $1-D$ XY model, the Hamiltonian presents three terms with ground states that exhibit different phases. The first two terms parametrized by $J$ and $\gamma$ are $\sigma_i^x\sigma_{i+1}^x$ and $\sigma_i^y\sigma_{i+1}^y$. Both by themselves correspond to the well-known Ising model, in which the ground state is ferromagnetic or antiferromagnetic, depending on the sign of $J$, and points respectively to the $x$ or $y$ axis. Contrarily, the ground state of the third term $\sigma_i^z$, parametrized by $\lambda$ is paramagnetic and points to the $z$ axis. As a result, the ground state will show ferromagnetic or antiferromagnetic behavior when $|J| > \lambda$ and the direction of the spin will be mediated by $\gamma$. However, the ground state will show paramagnetic behavior for $|J| < \lambda$. In Fig.1 there is shown the phase diagram at $T = 0$ of the 1-D XY model for $J = -1$.

In the next subsections, we derive the analytical solution of the XY model. However, before starting is convenient to rewrite Eq.(1) it in terms of spin ladder operators $\sigma^{+(-)}$ which increase(decrease) the projection of the third component of the spin $S_z$ by 1. The $\sigma^x$ and $\sigma^y$ operators then can be written as

$$\begin{aligned}\sigma^x &= \sigma^+ + \sigma^-,\\ \sigma^y &= -i\left(\sigma^+ + \sigma^-\right),\\ \sigma^z &= 2\sigma^+\sigma^- - 1.\end{aligned} \tag{2}$$

Hence, the Hamiltonian from Eq.(1) becomes

$$\mathcal{H}'_{XY} = J\left(\sum_{i=1}^{n-1} \sigma_i^+\sigma_{i+1}^- + \sigma_i^-\sigma_{i+1}^+ + \gamma\left(\sigma_i^+\sigma_{i+1}^+ + \sigma_i^-\sigma_{i+1}^-\right)\right) + \lambda\sum_{i=1}^{n}\left(2\sigma^+\sigma^- - 1\right). \tag{3}$$

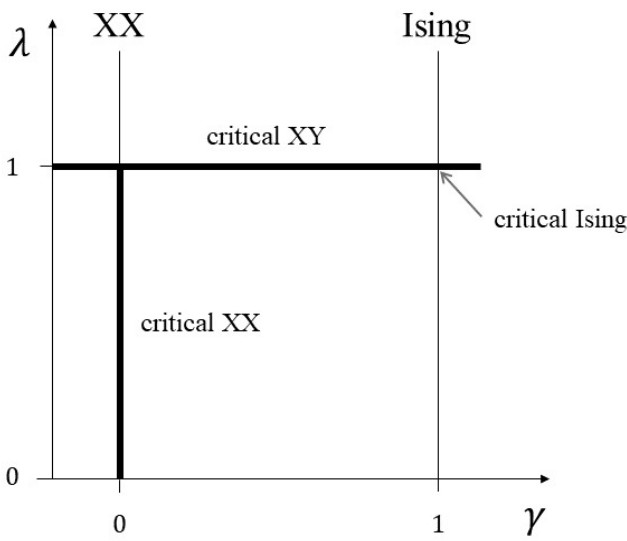

Figure 1: Phase diagram of the quantum XY model.

Furthermore, it is worth remembering some properties from the Spin $\frac{1}{2}$, which will be used later on in the next steps.

$$
\begin{aligned}
\sigma^+(-\sigma^z) &= \sigma^+, & \sigma^-(-\sigma^z) &= -\sigma^-, \\
-\sigma^z\sigma^+ &= -\sigma^+, & -\sigma^z\sigma^- &= \sigma^-.
\end{aligned}
\tag{4}
$$

## 2.1 Jordan-Wigner transformation

Generally, quantum spin objects are notoriously difficult to deal with in many-body physics because they neither fulfill fermionic nor bosonic algebra. For this reason, the first step to diagonalize XY Hamiltonian consists of applying the Jordan-Wigner transformation [19] which maps the spin operators $\sigma$ into spinless fermionic modes $c$.

The Jordan-Wigner transformation takes advantage of the similarities between fermions and spin operators. The existence of the stated similarity can be noticed by how both operators act on their respective basis, where fermionic basis $|1\rangle$ and $|0\rangle$ respectively corresponds to having one or no fermion in the state (no fermion state is also called void), while $|+\rangle$ and $|-\rangle$ means having a spin pointing up or down in the $z$ axis. As shown in Table 1, there is a clear equivalence between $|0\rangle$ and $|-\rangle$, and the same with $|1\rangle$ and $|+\rangle$.

Table 1: Fermionic and spin operator's behavior when acting in their respective basis.

| Fermions | Spin $\frac{1}{2}$ |
|---|---|
| $c^\dagger\|0\rangle = \|1\rangle$ | $\sigma^+\|-\rangle = \|+\rangle$ |
| $c^\dagger\|1\rangle = 0$ | $\sigma^+\|+\rangle = 0$ |
| $c\|0\rangle = 0$ | $\sigma^-\|-\rangle = 0$ |
| $c\|1\rangle = \|0\rangle$ | $\sigma^-\|+\rangle = \|-\rangle$ |

However, there is also an important difference between them, their commutation relationships. The commutation relationship followed by $\sigma^{+(-)}$ and $\sigma^-$ operators are

$$
\begin{aligned}
[\sigma_j^+, \sigma_i^-] &= 0, && i \neq j, \\
\{\sigma_i^+, \sigma_j^-\} &= I, && i = j,
\end{aligned}
\tag{5}
$$

while the operators $c$ and $c^\dagger$ obey the fermionic algebra $\{c_i^\dagger, c_j\} = \delta_{ij}$.

To solve this problem, Jordan and Wigner introduced an operator, called the string operator $e^{\pi i \sum_{j=1}^{i-1} \sigma_j^+ \sigma_j^-}$. This operator counts the number of $|+\rangle$ states or fermionic particles in the system and ensures the addition of a minus sign whenever two fermions are interchanged, thereby enabling the correct mapping between spin and fermionic operators.

Thus, the Jordan-Wigner transformation is defined as

$$
\begin{aligned}
c_i^\dagger &= \sigma_i^+ e^{-\pi i \sum_{j=1}^{i-1} \sigma_j^+ \sigma_j^-}, \\
c_i &= e^{\pi i \sum_{j=1}^{i-1} \sigma_j^+ \sigma_j^-} \sigma_j^-, \\
c_i^\dagger c_i &= \sigma_j^+ \sigma_j^-,
\end{aligned}
\tag{6}
$$

where the $c$ and $c_i^\dagger$ are the new spinless fermionic operators. Note that $\sigma_j^+ \sigma_j^-$ and $\sigma_i^+ \sigma_i^-$ commute

$$
[\sigma_i^+ \sigma_i^-, \sigma_j^-] = -\delta_{ij} \sigma_j^-, \qquad [\sigma_i^+ \sigma_i^-, \sigma_j^+] = \delta_{ij} \sigma_j^+, \qquad [\sigma_i^+ \sigma_i^-, \sigma_j^+ \sigma_j^-] = 0.
\tag{7}
$$

Therefore,

$$
e^{\pm i\pi \sum_{j=n}^{m} \sigma_j^+ \sigma_j^-} = \prod_{j=n}^{m} e^{\pm i\pi \sigma_j^+ \sigma_j^-}.
\tag{8}
$$

The next step is to develop the exponential operator

$$
e^{\pm i\pi \sigma_j^+ \sigma_j^-} = \sum_{l=0}^{\infty} \frac{1}{l!} (\pm i\pi)^l \left(\sigma_j^+ \sigma_j^-\right)^l = 1 - 2\sigma_j^+ \sigma_j^- = -\sigma_j^z.
\tag{9}
$$

In consequence, sometimes the Wigner-Jordan transformation is also written as

$$
c_i^\dagger = \sigma_i^+ \left(\prod_{l=1}^{i-1} -\sigma_l^z\right), \qquad c_i = \left(\prod_{l=1}^{i-1} -\sigma_l^z\right) \sigma_j^-, \qquad c_i^\dagger c_i = \sigma_j^+ \sigma_j^-.
\tag{10}
$$

For practical reasons, it is interesting to write down the inverse transformation

$$
\begin{aligned}
\sigma_i^+ &= c_i^\dagger \left(\prod_{l=1}^{i-1} -\sigma_l^z\right) = c_i^\dagger e^{\pi i \sum_{j=1}^{i-1} \sigma_j^+ \sigma_j^-}, \\
\sigma_i^- &= \left(\prod_{l=1}^{i-1} -\sigma_l^z\right) c_i = e^{-\pi i \sum_{j=1}^{i-1} \sigma_j^+ \sigma_j^-} c_i, \\
\sigma_i^z &= 2c_i^\dagger c_i - 1.
\end{aligned}
\tag{11}
$$

Now the transformed spin operators obey the canonical fermion algebra

$$
\{c_i, c_j^\dagger\} = \delta_{ij}, \qquad \{c_i, c_j\} = 0, \qquad \{c_i^\dagger, c_j^\dagger\} = 0,
\tag{12}
$$

as it is shown in Ref. [20].

Subsequently, let's derive some useful relations. Using Eq.(12) we can to compute the following commutator

$$[c_i^\dagger c_i, c_j^\dagger c_j] = 0 \,. \tag{13}$$

Additional useful commutator relations are

$$[c_i^\dagger c_i, c_j] = -\delta_{ji} c_i \,, \qquad [c_i^\dagger c_i, c_j^\dagger] = \delta_{ji} c_i^\dagger \,, \qquad (c_i^\dagger c_i)^2 = c_i^\dagger c_i \,. \tag{14}$$

Applying the different properties derived from the expressions before and $c_i c_i = 0$, one can compute

$$\{1 - 2c_i^\dagger c_i, c_i\} = 0 \,, \qquad \{1 - 2c_i^\dagger c_i, c_i^\dagger\} = 0 \,. \tag{15}$$

Now, using the properties from Eq.(14) and Eq.(15), one can compute the commutation relationship between the string operator described in Eq.(8) and the new fermionic operators

$$\begin{aligned}
[e^{\pm i\pi \sum_{j=n}^{m} c_j^\dagger c_j}, c_i] &= [e^{\pm i\pi \sum_{j=n}^{m} c_j^\dagger c_j}, c_i^\dagger] = 0 \,, & i \notin [n, m] \,, \\
\{e^{\pm i\pi \sum_{j=n}^{m} c_j^\dagger c_j}, c_i\} &= \{e^{\pm i\pi \sum_{j=n}^{m} c_j^\dagger c_j}, c_i^\dagger\} = 0 \,, & i \in [n, m] \,.
\end{aligned} \tag{16}$$

**Jordan-Wigner transformation in the XY model**

Here, we apply the Jordan-Wigner transformation into the elements that appear in the XY Hamiltonian, from Eq.(3). For the $\sigma_i^+ \sigma_{i+1}^+$ term,

$$\sigma_i^+ \sigma_{i+1}^+ = c_i^\dagger c_{i+1}^\dagger \,, \tag{17}$$

where we have taken in count that $c_i^\dagger c_i^\dagger = 0$. Likewise, one can calculate the $\sigma_i^- \sigma_{i+1}^-$ term

$$\sigma_i^- \sigma_{i+1}^- = c_{i+1} c_i \,. \tag{18}$$

Lastly, the $\sigma_i^+ \sigma_{i+1}^-$ and $\sigma_i^- \sigma_{i+1}^+$ can be transformed

$$\sigma_i^+ \sigma_{i+1}^- = c_i^\dagger c_{i+1} \,, \qquad \sigma_i^- \sigma_{i+1}^+ = c_{i+1}^\dagger c_i \,. \tag{19}$$

**Boundary conditions**

Until now, we have not mentioned anything about what happens in the boundary terms $\sigma_{n+1}$. Given the finite nature of our simulations, it becomes imperative to establish certain boundary conditions for our system. Specifically, we've implemented periodic boundary conditions (PBC). However, it's worth noting that we've opted for a direct application of PBC within the fermionic space. This choice translates to the relationship between fermionic operators, namely, $c_n c_{n+1} = c_n c_1$.

To add this term to our XY Hamiltonian, first, it has to be mapped into the spin space using the Jordan-Wigner transformation. Unfortunately, this transformation maps the PBC to PBC or antiperiodic boundary condition (APBC) depending on whether the system has an odd or even number of particles or $|+\rangle$ states. Consequently, the boundary term of our Hamiltonian must present this parity dependence to correctly be mapped into PBC in the fermionic space, this can be achieved using the $\sigma_1^y \sigma_2^z \cdots \sigma_{n-1}^z \sigma_n^y$ and $\sigma_1^x \sigma_2^z \cdots \sigma_{n-1}^z \sigma_n^x$ terms from Eq.(1). Then the Hamiltonian simulated in this work reads

$$\begin{aligned}
\mathcal{H}_{XY} = J &\left( \sum_{i=1}^{n-1} \frac{1+\gamma}{2} \sigma_i^x \sigma_{i+1}^x + \frac{1-\gamma}{2} \sigma_i^y \sigma_{i+1}^y \right) + \lambda \sum_{i=1}^{n} \sigma_i^z \\
&+ J \frac{1+\gamma}{2} \sigma_1^y \sigma_2^z \cdots \sigma_{n-1}^z \sigma_n^y + J \frac{1-\gamma}{2} \sigma_1^x \sigma_2^z \cdots \sigma_{n-1}^z \sigma_n^x \,.
\end{aligned} \tag{20}$$

The first two terms correspond to the $1-D$ XY Hamiltonian, whereas the last two terms belong to the boundary conditions. These boundary terms can be substituted with the conventional periodic terms $\sigma_n^x \sigma_1^x$ and $\sigma_n^y \sigma_1^y$ for states with an even number of spins pointing up, and the same terms with a negative sign for states with an odd number of spins pointing up. It is worth keeping in mind that even the Hamiltonian we are working on is not strictly the same as the XY model, in the thermodynamic limit the boundary conditions do not play any role and we recover the same results.

Now we will demonstrate that when the Jordan-Wigner transformation is applied to this term, the PBC is recovered for the fermionic operators. First, we need to write the $\sigma_1^y \sigma_2^z \cdots \sigma_{n-1}^z \sigma_n^y$ and $\sigma_1^x \sigma_2^z \cdots \sigma_{n-1}^z \sigma_n^x$ using the $\sigma^+$ and $\sigma^-$ operators

$$
\begin{aligned}
\sigma_1^y \sigma_2^z \cdots \sigma_{n-1}^z \sigma_n^y &= -\sigma_1^+ \sigma_2^z \cdots \sigma_{n-1}^z \sigma_n^+ + \sigma_1^+ \sigma_2^z \cdots \sigma_{n-1}^z \sigma_n^- \\
&\quad + \sigma_1^- \sigma_2^z \cdots \sigma_{n-1}^z \sigma_n^+ - \sigma_1^- \sigma_2^z \cdots \sigma_{n-1}^z \sigma_n^-, \\
\sigma_1^x \sigma_2^z \cdots \sigma_{n-1}^z \sigma_n^x &= \sigma_1^+ \sigma_2^z \cdots \sigma_{n-1}^z \sigma_n^+ + \sigma_1^+ \sigma_2^z \cdots \sigma_{n-1}^z \sigma_n^- \\
&\quad + \sigma_1^- \sigma_2^z \cdots \sigma_{n-1}^z \sigma_n^+ + \sigma_1^- \sigma_2^z \cdots \sigma_{n-1}^z \sigma_n^-.
\end{aligned}
\tag{21}
$$

Next, the Jordan-Wigner transformation is applied to the different terms that appear in the above expression using the properties shown in Eq.(4), $\sigma^z \sigma^z = 1$ and we will restrict our system to have an even number of qubits ($n$). First let's compute the term $\sigma_1^+ \sigma_2^z \cdots \sigma_{n-1}^z \sigma_n^+$,

$$
\sigma_1^+ \sigma_2^z \cdots \sigma_{n-1}^z \sigma_n^+ = c_1^\dagger \sigma_2^z \cdots \sigma_{n-1}^z \left( \prod_{l=1}^{n-1} -\sigma_l^z \right) c_n^\dagger = c_1^\dagger c_n^\dagger.
\tag{22}
$$

The rest of the terms can be computed following the same steps, the results are summarized in the following expressions

$$
\begin{aligned}
\sigma_1^+ \sigma_2^z \cdots \sigma_{n-1}^z \sigma_n^+ = c_1^\dagger c_n^\dagger, &\qquad \sigma_1^+ \sigma_2^z \cdots \sigma_{n-1}^z \sigma_n^- = c_1^\dagger c_n, \\
\sigma_1^- \sigma_2^z \cdots \sigma_{n-1}^z \sigma_n^+ = c_n^\dagger c_1, &\qquad \sigma_1^- \sigma_2^z \cdots \sigma_{n-1}^z \sigma_n^- = c_n c_1.
\end{aligned}
\tag{23}
$$

Subsequently, the boundary term reads

$$
\begin{aligned}
\mathcal{H}_{BC} &= \frac{1+\gamma}{2} \sigma_1^y \sigma_2^z \cdots \sigma_{n-1}^z \sigma_n^y + \frac{1-\gamma}{2} \sigma_1^x \sigma_2^z \cdots \sigma_{n-1}^z \sigma_n^x \\
&= c_1^\dagger c_n + c_n^\dagger c_1 + \gamma \left( c_n^\dagger c_1^\dagger + c_1 c_n \right) = c_{n+1}^\dagger c_n + c_n^\dagger c_{n+1} + \gamma \left( c_n^\dagger c_{n+1}^\dagger + c_{n+1} c_n \right),
\end{aligned}
\tag{24}
$$

where now is easy to see that this Hamiltonian fulfills the PBC in the fermionic space.

## 2.2 Fermionic Fourier transform (fFT)

Combining all the solutions outlined in the previous sections yields the Hamiltonian corresponding to the XY model but now is quadratic in fermionic annihilation and creation operators $c$ and $c^\dagger$ instead of quadratic in spin operator $\sigma^+$ and $\sigma^-$

$$
\mathcal{H}_{JW} = J \sum_{i=1}^{n} \left( c_i^\dagger c_{i+1} + c_{i+1}^\dagger c_i + \gamma \left( c_i^\dagger c_{i+1}^\dagger + c_{i+1} c_i \right) \right) + \lambda \sum_{i=1}^{n} \left( 2 c_i^\dagger c_i - 1 \right).
\tag{25}
$$

Hamiltonians that are quadratic in fermionic creation and annihilation operators are ubiquitous in condensed matter physics. They describe systems of non-interacting fermions and also arise in the mean-field treatment of more complex interacting systems. Diagonalizing such Hamiltonians is a well-established procedure, typically accomplished using spatially dependent couplings and techniques such as the Bogoliubov transformation [21, 22] and fermionic Gaussian states [23]. In translationally invariant models, such as the XY model, the Fourier

transform is particularly useful, as it partially diagonalizes the Hamiltonian by making it local in momentum space. However, this transformation often introduces anomalous terms that couple different momentum modes, thus requiring a subsequent Bogoliubov transformation to achieve full diagonalization.

In the second quantization, the Fourier transform is defined as

$$c_j = \frac{1}{\sqrt{N}} \sum_{k=-\frac{n}{2}+1}^{\frac{n}{2}} b_k e^{i\frac{2\pi k}{n}j}, \qquad c_j^\dagger = \frac{1}{\sqrt{N}} \sum_{k=-\frac{n}{2}+1}^{\frac{n}{2}} b_k^\dagger e^{-i\frac{2\pi k}{n}j}, \tag{26}$$

where $b_k^\dagger$ and $b_k$ are the creation and annihilation operators of the fermionic Fourier modes.

The discrete $k$ values are acquired establishing the translational invariance of the system by PBC

$$|x+n\rangle = |x\rangle,$$
$$\sum_k e^{i\frac{2\pi k}{n}(x+n)} |k\rangle = \sum_{k'} e^{i\frac{2\pi k'}{n}(x)} |k'\rangle. \tag{27}$$

Then, we multiply at both sides by $\langle k|$, and applying $\langle k|k'\rangle = \delta_{k,k'}$

$$e^{i\frac{2\pi k}{n}(x+n)} = e^{i\frac{2\pi k}{n}(x)} \qquad \rightarrow \qquad e^{i\frac{2\pi kn}{n}} = 1,$$
$$\frac{2\pi kn}{n} = 2\pi m \qquad \rightarrow \qquad k = m, \tag{28}$$

where $m$ is an integer. Because the number of qubits ($n$) is even, as mentioned in Section 2.1, we can choose our $k$ values to be

$$k = -\frac{n}{2}+1, -\frac{n}{2}+2, \ldots, -1, 0, 1, \ldots, \frac{n}{2}-1, \frac{n}{2}. \tag{29}$$

In the case where the number of qubits is odd, from Eq.(22) it can be seen that an extra "−" sign appears in the final result obtaining APBC $-c_1^\dagger c_n^\dagger$. Then applying translational invariance we get that the $k$ possible values are the same as in the previous case. As surprising as it may seem, one can expect this result if one thinks in terms of sinusoidal functions. If the period of the sinusoidal function is $L$ we will recover in $x = 0$ the same result as in $x = L$, hence we have PBC. Nonetheless, if our lattice ends in $x = \frac{L}{2}$ then we will have the same absolute value in $x = 0$ and $x = \frac{L}{2}$ but with a different sign. As a result, we have APBC. In the end, the $n$ odd case for APBC must have the same $k$ values as $2n$ with PBC.

Even though we will be only focusing on the even number of qubits case, the procedure followed for the odd case will be equivalent to the one we will describe for the even case. One can find more information about the general case and boundary conditions in Ref. [24].

**Fermionic Fourier transform in the XY model**

Before computing the new terms of the XY Hamiltonian, let us first recall the fundamental properties of the FT

$$\frac{1}{N} \sum_k e^{i\frac{2\pi k}{n}(j-j')} = \delta_{j,j'}, \qquad \frac{1}{N} \sum_j e^{i\frac{2\pi (k-q)}{n}j} = \delta_{k,q}, \tag{30}$$

where $\delta_{j,j'}$ and $\delta_{k,q}$ are Kronecker deltas, which are 0 when $j \neq j'$ or $k \neq q$ and 1 when are equals.

By applying Eq.(30) to each term appearing in Eq.(25), we obtain the FT of our Hamiltonian,

$$\sum_{j=1}^{n} c_j^\dagger c_{j+1} = \sum_{j=1}^{n} \left( \frac{1}{\sqrt{n}} \sum_{k=-\frac{n}{2}+1}^{\frac{n}{2}} b_k^\dagger e^{-i\frac{2\pi k}{n}j} \right) \left( \frac{1}{\sqrt{n}} \sum_{k'=-\frac{n}{2}+1}^{\frac{n}{2}} b_{k'} e^{i\frac{2\pi k'}{n}(j+1)} \right) = \sum_k b_k^\dagger b_k e^{i\frac{2\pi k}{n}}, \quad (31)$$

$$\sum_{j=1}^{n} c_{j+1} c_j = \sum_{j=1}^{n} \left( \frac{1}{\sqrt{n}} \sum_k b_k e^{i\frac{2\pi k}{n}(j+1)} \right) \left( \frac{1}{\sqrt{n}} \sum_{k'} b_{k'} e^{i\frac{2\pi k'}{n}j} \right) = \sum_k i \sin\left( \frac{2\pi k}{n} \right) b_k b_{-k}, \quad (32)$$

$$\sum_{j=1}^{n} c_j^\dagger c_{j+1}^\dagger = \sum_{j=1}^{n} \left( \frac{1}{\sqrt{n}} \sum_k b_k^\dagger e^{-i\frac{2\pi k}{n}j} \right) \left( \frac{1}{\sqrt{n}} \sum_{k'} b_{k'}^\dagger e^{-i\frac{2\pi k'}{n}(j+1)} \right) = \sum_k i \sin\left( \frac{2\pi k}{n} \right) b_k^\dagger b_{-k}^\dagger, \quad (33)$$

$$\sum_{j=1}^{n} c_{j+1}^\dagger c_j = \sum_{j=1}^{N} \left( \frac{1}{\sqrt{n}} \sum_k b_k^\dagger e^{-i\frac{2\pi k}{n}(j+1)} \right) \left( \frac{1}{\sqrt{n}} \sum_{k'} b_{k'} e^{i\frac{2\pi k'}{n}j} \right) = \sum_k b_k^\dagger b_k e^{-i\frac{2\pi k}{n}}, \quad (34)$$

$$\sum_{j=1}^{n} c_j^\dagger c_j = \sum_{j=1}^{N} \left( \frac{1}{\sqrt{n}} \sum_k b_k^\dagger e^{-i\frac{2\pi k}{n}(j)} \right) \left( \frac{1}{\sqrt{n}} \sum_{k'} b_{k'} e^{i\frac{2\pi k'}{n}j} \right) = \sum_k b_k^\dagger b_k. \quad (35)$$

The transformed Hamiltonian becomes

$$\mathcal{H}_{FT} = \sum_k \left[ 2\left( \lambda + J \cos\left( \frac{2\pi k}{n} \right) \right) b_k^\dagger b_k + iJ\gamma \sin\left( \frac{2\pi k}{n} \right) (b_k^\dagger b_{-k}^\dagger + b_k b_{-k}) \right] - \lambda n. \quad (36)$$

As a result of working in momentum space, the XY-Hamiltonian does not contain mixed terms between first neighbors, however, it is not diagonal yet because it contains terms with opposite momentum $k$ and $-k$ coupled.

For future calculations, it is beneficial to rewrite the Eq.(36) making use of the cosine function parity $(\cos(\alpha) = \cos(-\alpha))$, acknowledging that the summation takes over positive and negative $k$ values and without carrying the constant term $\lambda n$. Thereafter, the Hamiltonian is expressed as follows

$$\mathcal{H}'_{FT} = \sum_k \left[ \left( \lambda + J \cos\left( \frac{2\pi k}{n} \right) \right) (b_k^\dagger b_k + b_{-k}^\dagger b_{-k}) + iJ\gamma \sin\left( \frac{2\pi k}{n} \right) (b_k^\dagger b_{-k}^\dagger + b_k b_{-k}) \right]$$
$$= \sum_k \left[ \epsilon_k \left( b_k^\dagger b_k - b_{-k} b_{-k}^\dagger + 1 \right) + i\Delta_k (b_k^\dagger b_{-k}^\dagger + b_k b_{-k}) \right]. \quad (37)$$

The last term can be rewritten in matrix-vector form, then the expression becomes

$$\sum_k \begin{pmatrix} b_k^\dagger & b_{-k} \end{pmatrix} \begin{pmatrix} \epsilon_k & i\Delta_k \\ -i\Delta_k & -\epsilon_k \end{pmatrix} \begin{pmatrix} b_k \\ b_{-k}^\dagger \end{pmatrix} + \sum_k \epsilon_k. \quad (38)$$

Here, the definitions of $\epsilon_k = \lambda + J \cos\left( \frac{2\pi k}{n} \right)$ and $\Delta_k = J\gamma \sin\left( \frac{2\pi k}{n} \right)$ serve the purpose of enhancing the clarity of the upcoming mathematical development.

## 2.3 Bogoliubov transformation

The last step to diagonalize the Hamiltonian completely is the Bogoliubov transformation. This transformation is used to diagonalize quadratic Hamiltonians, for instance, it is used in the Superconductivity BSC theory or solid-state physics in Hamiltonians described by phononic interactions [25]. It can be understood as a change of basis, where the new base decouples the opposite momentum terms.

Specifically, the transformation will have a form such as

$$a_k = u_k b_k + v_k b_{-k}^\dagger, \qquad a_{-k} = u_{-k} b_{-k} + v_{-k} b_k^\dagger,$$
$$a_k^\dagger = u_k^* b_k^\dagger + v_k^* b_{-k}, \qquad a_{-k}^\dagger = u_{-k}^* b_{-k}^\dagger + v_{-k}^* b_k, \tag{39}$$

where $a_k^\dagger$ and $a_k$ are the Bogoulibov fermionic annihilation and creation operators associated with pseudo-momentum $k$, while $a_{-k}^\dagger$ and $a_{-k}$ are the Bogoulibov fermionic annihilation and creation operators associated with pseudo-momentum $-k$.

Because we are working in a fermionic system, we have to impose the anticommutation relationship of these new operators

$$\{a_k, a_k^\dagger\} = 1 \qquad \rightarrow \qquad |u_k|^2 + |v_k|^2 = 1,$$
$$\{a_k, a_{-k}\} = 0 \qquad \rightarrow \qquad u_k v_{-k} + v_k u_{-k} = 0. \tag{40}$$

To fulfill the second relationship, we use the condition $v_{-k} = -v_k$. This last condition along with Eq.(39) could be used to reverse the fermionic operator transformation. The old fermionic operators as a linear combination of the new fermionic operators are

$$b_k = u_k^* a_k - v_k a_{-k}^\dagger, \qquad b_{-k} = u_k^* a_{-k} + v_k a_k^\dagger,$$
$$b_k^\dagger = u_k a_k^\dagger - v_k^* a_{-k}, \qquad b_{-k}^\dagger = u_k a_{-k}^\dagger + v_k^* a_k. \tag{41}$$

For our purposes, it is useful to arrange the last expression in the vector-matrix form

$$\begin{pmatrix} b_k \\ b_{-k}^\dagger \end{pmatrix} = \begin{pmatrix} u_k^* & -v_k \\ v_k^* & u_k \end{pmatrix} \begin{pmatrix} a_k \\ a_{-k}^\dagger \end{pmatrix}. \tag{42}$$

The next step consists of passing from a non-diagonal Hamiltonian $\mathcal{H}_{FT}$ to a diagonal one by applying a change of basis matrix, which transforms the $b_k$ to $a_k$ operators.

$$\mathcal{H}'_{Bog} = \sum_k \begin{pmatrix} a_k^\dagger & a_{-k} \end{pmatrix} \begin{pmatrix} u_k & v_k \\ -v_k^* & u_k^* \end{pmatrix} \begin{pmatrix} \epsilon_k & i\Delta_k \\ -i\Delta_k & -\epsilon_k \end{pmatrix} \begin{pmatrix} u_k^* & -v_k \\ v_k^* & u_k \end{pmatrix} \begin{pmatrix} a_k \\ a_{-k}^\dagger \end{pmatrix}. \tag{43}$$

The Hamiltonian matrix written in terms of $a_k$ operators becomes

$$\begin{pmatrix} \epsilon_k \left(|u_k|^2 - |v_k|^2\right) + i\Delta_k \left(u_k v_k^* - u_k^* v_k\right) & -2\epsilon_k u_k v_k + i\Delta_k \left(u_k u_k + v_k v_k\right) \\ -2\epsilon_k u_k^* v_k^* - i\Delta_k \left(u_k^* u_k^* + v_k^* v_k^*\right) & -\left(\epsilon_k \left(|u_k|^2 - |v_k|^2\right) + i\Delta_k \left(u_k v_k^* - u_k^* v_k\right)\right) \end{pmatrix}. \tag{44}$$

The Bogoliubov modes that diagonalize the Hamiltonian are found by vanishing the non-diagonal terms. For this purpose, it is convenient to express $u_k$ and $v_k$ as

$$u_k = e^{\phi_1} \cos\left(\frac{\theta_k}{2}\right), \qquad v_k = e^{\phi_2} \sin\left(\frac{\theta_k}{2}\right). \tag{45}$$

Substituting the last expression in the non-diagonal term of Eq.(44) and making it vanish, one gets the expression

$$-2\epsilon_k e^{\phi_1 + \phi_2} \cos\left(\frac{\theta_k}{2}\right) \sin\left(\frac{\theta_k}{2}\right) + i\Delta_k u_k u_k - (-i\Delta_k) v_k v_k = 0 \qquad \rightarrow$$
$$-2\epsilon_k e^{\phi_1 + \phi_2} \cos\left(\frac{\theta_k}{2}\right) \sin\left(\frac{\theta_k}{2}\right) + \Delta_k e^{2\phi_1 + \frac{\pi}{2}} \cos^2\left(\frac{\theta_k}{2}\right) - \Delta_k e^{2\phi_2 - \frac{\pi}{2}} \sin^2\left(\frac{\theta_k}{2}\right) = 0. \tag{46}$$

If one wishes to vanish the phase term in the expression, the relation $\phi_1 + \phi_2 = 2\phi_1 + \frac{\pi}{2} = 2\phi_2 - \frac{\pi}{2}$ must be fulfilled. Without loss of generality, the relative phase can be chosen as $\phi_1 = 0$ and $\phi_1 = \frac{\pi}{2}$. Accordingly, the new fermionic operators $a_k^\dagger$ and $a_k$ are

$$
\begin{aligned}
a_k &= \cos\left(\frac{\theta_k}{2}\right) b_k + i \sin\left(\frac{\theta_k}{2}\right) b_{-k}^\dagger, & a_{-k} &= \cos\left(\frac{\theta_k}{2}\right) b_{-k} - i \sin\left(\frac{\theta_k}{2}\right) b_k^\dagger, \\
a_k^\dagger &= \cos\left(\frac{\theta_k}{2}\right) b_k^\dagger - i \sin\left(\frac{\theta_k}{2}\right) b_{-k}, & a_{-k}^\dagger &= \cos\left(\frac{\theta_k}{2}\right) b_{-k}^\dagger + i \sin\left(\frac{\theta_k}{2}\right) b_k.
\end{aligned}
\tag{47}
$$

In addition, using the expressions $\sin(2\theta) = 2\cos(\theta)\sin(\theta)$ and $\cos(2\theta) = \cos^2(\theta) - \sin^2(\theta)$, the Eq.(46) becomes

$$
\tan(\theta_k) = \frac{\Delta_k}{\epsilon_k}. \tag{48}
$$

It is now possible to obtain the required expressions to compute the diagonal energy terms $(E_k)$

$$
\begin{aligned}
|u_k|^2 - |v_k|^2 &= \cos^2\left(\frac{\theta_k}{2}\right) - \sin^2\left(\frac{\theta_k}{2}\right) = \cos(\theta_k) = \frac{1}{\sqrt{1 + \tan(\theta_k)}} = \frac{\epsilon_k}{\sqrt{\epsilon_k^2 + \Delta_k^2}}, \\
u_k v_k &= u_k^* v_k = \frac{i}{2}\sin(\theta_k) = \frac{i}{2}\tan(\theta_k)\cos(\theta_k) = \frac{i}{2}\frac{\Delta_k}{\sqrt{\epsilon_k^2 + \Delta_k^2}}, \\
u_k v_k^* &= u_k^* v_k^* = -\frac{i}{2}\frac{\Delta_k}{\sqrt{\epsilon_k^2 + \Delta_k^2}}, \\
E_k &= \sqrt{\epsilon_k^2 + \Delta_k^2}.
\end{aligned}
\tag{49}
$$

Therefore, Eq.(43) has the diagonal form

$$
\begin{aligned}
\mathcal{H}'_{Bog} &= \sum_k \begin{pmatrix} a_k^\dagger & a_{-k} \end{pmatrix} \begin{pmatrix} E_k & 0 \\ 0 & -E_k \end{pmatrix} \begin{pmatrix} a_k \\ a_{-k}^\dagger \end{pmatrix} = \sum_k E_k a_k^\dagger a_k - E_k a_{-k} a_{-k}^\dagger \\
&= \sum_k E_k \left( a_k^\dagger a_k + a_{-k}^\dagger a_{-k} - 1 \right) = \sum_{k=\frac{-n}{2}+1}^{\frac{n}{2}} 2 E_k \left( a_k^\dagger a_k - \frac{1}{2} \right).
\end{aligned}
\tag{50}
$$

Finally, the diagonal Hamiltonian has the form

$$
\tilde{\mathcal{H}} = \sum_{k=\frac{-n}{2}+1}^{\frac{n}{2}} \left[ 2 E_k \left( a_k^\dagger a_k - \frac{1}{2} \right) + \epsilon_k - \lambda \right], \tag{51}
$$

where $E_k = \sqrt{\left(\lambda + J\cos\left(\frac{2\pi k}{n}\right)\right)^2 + \left(J\gamma\sin\left(\frac{2\pi k}{n}\right)\right)^2}$ are the energies related to having one fermion in the Bogoulibov mode $k$ or $-k$. As a result, we have diagonalized the XY Hamiltonian.

## 3 Quantum circuit to diagonalize the XY model

In this section, we introduce a circuit $\mathcal{U}_{\mathrm{dis}}$ designed to convert the XY Hamiltonian $\mathcal{H}_{\mathrm{XY}}$ into its diagonal form $\bar{\mathcal{H}}_{\mathrm{XY}}$, by

$$
\bar{\mathcal{H}}_{\mathrm{XY}} = \mathcal{U}_{\mathrm{dis}} \mathcal{H}_{\mathrm{XY}} \mathcal{U}_{\mathrm{dis}}^\dagger. \tag{52}
$$

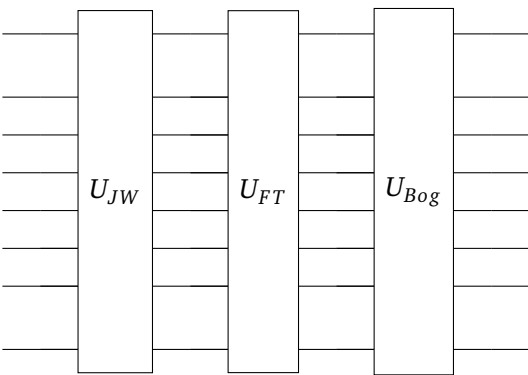

Figure 2: Schematic representation of the disentangling quantum circuit $U_{dis}$ for $n = 8$ qubits.

Using this transformation, we can obtain all eigenstates and any superposition of them in the spin basis, by preparing a product state in the computational basis and applying $\mathcal{U}_{\text{dis}}^{\dagger}$,

$$|XY \text{ eigenstate}\rangle = \mathcal{U}_{\text{dis}}^{\dagger} |\text{Comp. basis}\rangle . \tag{53}$$

Furthermore, we can reverse this process. Applying $\mathcal{U}_{\text{dis}}$ to states in the computational basis allows us to obtain any spin state represented in the diagonal basis.

Unfortunately, constructing these disentangling circuits $\mathcal{U}_{dis}$ for an arbitrary Hamiltonian is a challenging task. However, for models that present analytical solutions, we can try to map each step into a quantum operation. For the case it concerns us, the XY Hamiltonian needs three operations: $i$) Jordan-Wigner transformation, $ii$) Fourier transform, $iii$) Bogoliubov transformation. In the end, the disentangling circuit will exhibit the structure

$$\mathcal{U}_{dis} = \mathcal{U}_{Bog}.\mathcal{U}_{FT}\mathcal{U}_{JW} . \tag{54}$$

In the following section, we detail the construction of each $\mathcal{U}_{dis}$ operation using basic quantum gates.

## 3.1 Jordan-Wigner circuit

The Jordan-Wigner transformation maps the spin states to a fermionic spinless mode. In terms of the wave function,

$$|\Psi\rangle = \sum_{i_1,\ldots,i_n=0,1} \Psi_{i_1,\ldots,i_n} |i_1,\ldots,i_n\rangle = \sum_{i_1,\ldots,i_n=0,1} \Psi_{i_1,\ldots,i_n} \left(c_1^{\dagger}\right)^{i_1} \cdots \left(c_n^{\dagger}\right)^{i_n} |0\rangle , \tag{55}$$

where $i_j$ represent the state $i$ of the qubit at position $j$, with $j$ going from 1 to the number of qubits $n$. In spin and fermionic space, the $i$ can take values 0 or 1. In spin space, $|0\rangle = |+\rangle$ and $|1\rangle = |-\rangle$, while in fermionic space $|0\rangle_j$ means the $j$-th position is not occupied by a fermion, and $|1\rangle_j$ means having one fermion.

Notice that the coefficients $\Psi_{i_1,\ldots,i_n}$ remain unchanged under the transformation. Thus, in theory, no additional gate is required to implement the Jordan-Wigner transformation. However, two important caveats must be addressed here.

The first one arises when two-qubit states are exchanged using a SWAP operation. Since we are dealing with fermions, exchanging two fermions requires introducing a minus sign to

account for their antisymmetric nature. This adjustment is implemented using the fermionic SWAP operation (fSWAP). In matrix form, the fSWAP is represented as

$$fSWAP = \begin{pmatrix} 1 & 0 & 0 & 0 \\ 0 & 0 & 1 & 0 \\ 0 & 1 & 0 & 0 \\ 0 & 0 & 0 & -1 \end{pmatrix}, \tag{56}$$

which can be decomposed into a standard SWAP gate followed by a controlled-Z gate.

A second issue concerns a discrepancy in notation. In quantum computing, the spin states that are eigenstates of $\sigma_z$ with positive and negative eigenvalues are conventionally denoted as $|\uparrow\rangle = |0\rangle$ and $|\downarrow\rangle = |1\rangle$, respectively. In contrast, in many-body physics, the symbol $|0\rangle$ (or sometimes $|\Omega\rangle$) typically denotes the vacuum state. Since the Jordan-Wigner maps $|\downarrow\rangle$ into $|\Omega\rangle$, an $X$ gate has been introduced to keep the standard convention and avoid potential confusion. As a result, the circuit is initialized with a layer of $X$ gates applied to each qubit.

We want to stress that this decision is primarily for consistency with established conventions. Choosing not to apply $X$ gates is a valid alternative. In such a case, the unitary transformation required to disentangle the $XY$ model will differ slightly from the approach described in this work. However, the final result should remain unchanged.

## 3.2 Fermionic Fourier transform circuit

The next step involves transforming the fermionic modes into momentum space using the Fourier transform. When the number of particles is a power of two, meaning $n = 2^m$ where $m$ is a natural number, the fermionic Fourier transform can be implemented by following the classical fast Fourier transform scheme [26].

The idea is based on the work of Andrew J. Ferrys in Ref. [27]. First, we decompose the n-qubit Fourier transform in two parallel $\frac{n}{2}$-qubit Fourier transforms, one acting upon odd and even modes respectively

$$\begin{aligned} b_k^\dagger &= \frac{1}{\sqrt{n}} \sum_{j=0}^{n-1} e^{i\frac{2\pi}{n}jk} c_j^\dagger = \frac{1}{\sqrt{\frac{n}{2}2}} \sum_{j'=0}^{n/2-1} e^{i\frac{2\pi}{n}2j'k} c_{2j'}^\dagger + \frac{1}{\sqrt{\frac{n}{2}2}} \sum_{j'=0}^{n/2-1} e^{i\frac{2\pi}{n}(2j'+1)k} c_{2j'+1}^\dagger \\ &= \frac{1}{\sqrt{2}} \left[ \frac{1}{\sqrt{\frac{n}{2}}} \sum_{j'=0}^{n/2-1} e^{i\frac{2\pi}{n/2}j'k} c_{2j'}^\dagger + e^{i\frac{2\pi}{n}k} \frac{1}{\sqrt{\frac{n}{2}}} \sum_{j'=0}^{n/2-1} e^{i\frac{2\pi}{n/2}j'k} c_{2j'+1}^\dagger \right]. \end{aligned} \tag{57}$$

To avoid confusion with the operators defined earlier, we have chosen to use the tilde symbol to denote the operators derived from the FT in this section. In this context, $b_k^\dagger$ is equivalent to the operator $b_k^\dagger$ defined in Sec.2.2.

We can now define a new set of fermionic operators for even and odd sites $a_j \equiv c_{2j'}$ and $d_j \equiv c_{2j'+1}$. The fermionic Fourier transform of those operators using $\frac{n}{2}$ points will be

$$\tilde{a}_k^\dagger = \frac{1}{\sqrt{\frac{n}{2}}} \sum_{j=0}^{\frac{n}{2}-1} e^{i\frac{2\pi}{n}jk} a_j^\dagger, \qquad \tilde{d}_k^\dagger = \frac{1}{\sqrt{\frac{n}{2}}} \sum_{j=0}^{\frac{n}{2}-1} e^{i\frac{2\pi}{n}jk} d_j^\dagger. \tag{58}$$

If we now insert the prior definition in Eq.(57)

$$b_k^\dagger = \frac{1}{\sqrt{2}} \left[ \tilde{a}_k^\dagger + e^{i\frac{2\pi}{n}k} \tilde{d}_k^\dagger \right], \qquad b_{k+\frac{n}{2}}^\dagger = \frac{1}{\sqrt{2}} \left[ \tilde{a}_k^\dagger - e^{i\frac{2\pi}{n}k} \tilde{d}_k^\dagger \right], \tag{59}$$

where in the last equality we have used the periodicity of the Fourier transform. In the case of $\frac{n}{2}$ Fourier transform the period for $k$ values is $\frac{n}{2}$, so $\tilde{a}_{k+\frac{n}{2}}^\dagger = \tilde{a}_k^\dagger$ and exactly the same for $\tilde{d}_k^\dagger$ operator.

Equation (59) shows us that we can obtain the values of the $n$ qubit Fourier transform ($b_k$) from a $\frac{n}{2}$ ($a_k, d_k$) qubit Fourier transform. In the case of systems with $n = 2^m$ qubits, this process of division can continue iteratively until the Fourier transform is reduced to a 2-qubit operation. Notably, the 2-qubit Fourier transform has the same expression as Eq.(59) with $k = 0$.

At this stage, we have established the interplay between $b_k$ and their counterparts $a_k$ and $d_k$. Nevertheless, our primary goal is to derive the matrix that defines the relationship between $|k\rangle_b |k + \frac{n}{2}\rangle_b$ and $|k\rangle_a |k\rangle_d$ states. This matrix can be determined by recognizing that the vacuum state remains unchanged under the transformation, as the Fourier transform does not mix annihilation and creation operators in its definition. Hence, the remaining states can be attained by applying the creation operators to the void state, explicitly

- void vector (0 fermions)   $|0\rangle_{k_b} |0\rangle_{k_b + \frac{n}{2}} = |0\rangle_{k_a} |0\rangle_{k_d}$ ,

- apply $b^\dagger_{k+\frac{n}{2}}$ to obtain $|0\rangle_{k_b} |1\rangle_{k_b + \frac{n}{2}} = \frac{1}{\sqrt{2}} \left[ |1\rangle_{k_a} |0\rangle_{k_d} - e^{i\frac{2\pi}{n}k} |0\rangle_{k_a} |1\rangle_{k_d} \right]$ ,

- apply $b^\dagger_k$ to obtain $|1\rangle_{k_b} |0\rangle_{k_b + \frac{n}{2}} = \frac{1}{\sqrt{2}} \left[ |1\rangle_{k_a} |0\rangle_{k_d} + e^{i\frac{2\pi}{n}k} |0\rangle_{k_a} |1\rangle_{k_b} \right]$ ,

- apply $b^\dagger_k b^\dagger_{k+\frac{n}{2}}$ to obtain $|1\rangle_{k_b} |1\rangle_{k_b + \frac{n}{2}} = -e^{i\frac{2\pi}{n}k} |1\rangle_{k_a} |1\rangle_{k_d}$ .

Here, the subscript $k_b$ means that this vector belongs to the $n$-qubit Fourier space, $k_a$ indicates that the vector is associated with the $n$ even-qubit Fourier space, and $k_d$ denotes that the vector belongs to the $n$ odd-qubit Fourier space.

Deriving the matrix that performs the mentioned operation is a straightforward process. For the remainder of this work, we will refer to this matrix as the "General FT 2-qubit gate" or $F^n_k$. It takes the following form

$$
F^n_k \equiv \begin{pmatrix} 1 & 0 & 0 & 0 \\ 0 & \frac{-e^{-i\frac{2\pi}{n}k}}{2} & \frac{1}{\sqrt{2}} & 0 \\ 0 & \frac{e^{-i\frac{2\pi}{n}k}}{\sqrt{2}} & \frac{1}{\sqrt{2}} & 0 \\ 0 & 0 & 0 & -e^{-i\frac{2\pi}{n}k} \end{pmatrix} ,
\tag{60}
$$

where the $F^n_k$ matrix transforms the $|k\rangle_a |k\rangle_b$ vectors into $|k\rangle_c |k + \frac{n}{2}\rangle_c$. Additionally, the 2-qubit Fourier transform is recovered whenever $k = 0$, which is represented as $F_2$.

It is key to bear in mind that there is a gap between the gates that can be applied theoretically in a quantum computer and those that can currently be implemented on real devices. Therefore, all gates must be decomposed into basic gates that can be implemented in a quantum computer. In certain cases, analytical schemes exist for such decompositions [28]. For the case $F^n_k$, the decomposition into basic gates is shown in Fig.3.

Up to this point, we have found the 2-qubit gate transform. Now we will describe the circuit needed to perform the FT of $n$ qubits. The approach involves decomposing the $n$ qubits into the even and odd sectors, followed by applying the fermionic Fourier transform of $\frac{n}{2}$ qubits.

Once the $\frac{n}{2}$ Fourier transform is complete, the $F^n_k$ gate is applied to the $i$ qubit and the $i + \frac{n}{2}$ qubit. This process is repeated iteratively until the FT is reduced to 2-qubit operations, which will be performed by $F_2$. Nevertheless, depending on the connectivity of the qubits, additional fSWAP operations may be required. In this work, we assume a linear connectivity model, where qubits are arranged in a 1D configuration.

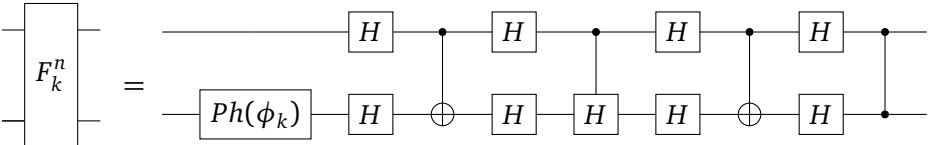

Figure 3: The diagram illustrates the decomposition of the building block of $F_k^n$ (Eq.(60)), where $\phi_k = \frac{-i2\pi k}{n}$.

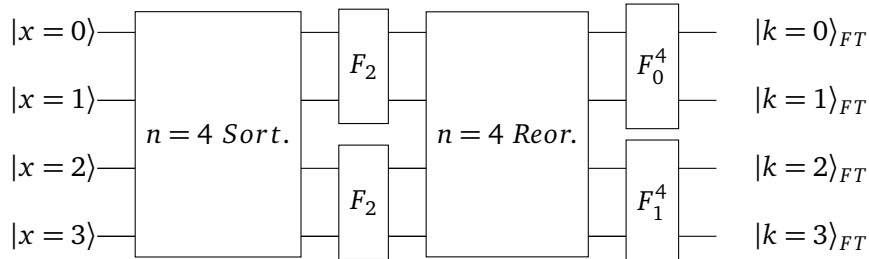

Figure 4: Scheme followed to perform the fermionic Fourier transform (fFT) for the case of $n = 4$ qubits. The first step of the algorithm corresponds to the qubit sorting (Sort.), then the fermionic Fourier transform for $n = 2$ ($F_2$) qubits is applied and performed into the even and odd sectors. The next step is the Fourier states reorganization (Reor.) and finally, the General Fourier transform 2-qubit gate ($F_k^n$) to recover the $k$ and $k+2$ states.

Next, we will describe the algorithm used to construct the fermionic Fourier transform circuit for $n$ qubits assuming linear connectivity and that the first qubit is numbered as 0. This circuit is decomposed into four phases:

1. **Qubit sorting (Sort.):** In the initial step, we categorize the qubits into even and odd sectors using fermionic SWAP gates whenever we exchange two qubits.

2. $\frac{n}{2}$ **Fermionic Fourier transform (fFT):** The second phase entails the application of the Fermionic Fourier Transform circuit for $\frac{n}{2}$ qubits into the even and odd sectors.

3. **Fourier states reorganization (Reor.):** Subsequently, we undertake the reordering of the resulting states to group the $k_{even}$ and $k_{odd}$ states.

4. **General Fourier transform gate application ($F_k^n$):** The final phase involves the application of the general Fourier transform gate $F_k^n$ to the $k_{even}$ and $k_{odd}$ states. This step is performed to recover the $k$ and $k + \frac{n}{2}$ states.

Figure 4 and Fig.5 represent the diagram of the fermionic Fourier transform for the case of $n = 4$ and $n = 8$ qubits respectively. Both pictures show the different parts of the algorithm described above.

**Qubit sorting**

The initial step involves the segregation of qubits into even and odd sectors through a series of fermionic SWAP operations, a process that occurs throughout $\frac{n}{2} - 1$ layers. In the first layer, precisely $\frac{n}{2} - 1$ fermionic gates come into play, each consecutively applied, starting with qubit 1. Subsequently, in each successive layer, one fewer gate is used than in the previous layer,

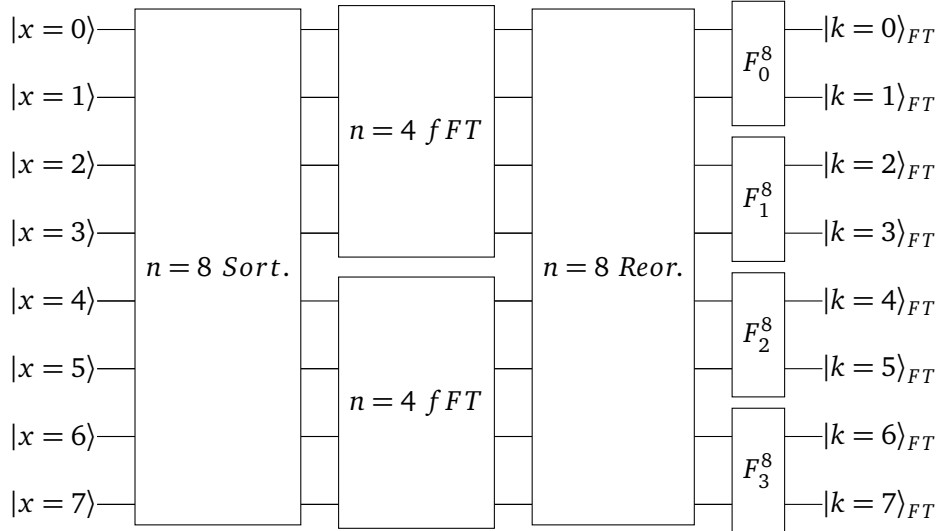

Figure 5: Scheme followed to perform the fermionic Fourier transform (fFT) for the case of $n = 8$ qubits. The first step of the algorithm corresponds to the qubit sorting (Sort.), then the fermionic Fourier transform for $n = 4$ (fFT) qubits is applied and performed into the even and odd sectors. The next step is the Fourier states reorganization (Reor.) and finally the general Fourier transform to recover the $k$ and $k + 4$ states.

following a sequential progression starting with the next qubit after the initial qubit of the preceding layer. Moreover, in Algorithm 1, we presented the algorithm in pseudocode:

---

**Algorithm 1** Qubit sorting circuit

---

**Require:** $num\_qubit = 2^m$

**Ensure:** $qc\_sorting \rightarrow$ Quantum circuit which separates the qubits in even and odd sectors.

  $num\_label = \frac{n}{2} - 1$

  $num\_gates = \frac{n}{2} - 1$

  $qubit\_init = 1$

  **for** $i = 1$ to $num\_label$ **do**

    $count\_qubit = qubit\_init$

    **for** j=$num\_gates$ to 1 **do**

      add fSWAP into $count\_qubit$ and $count\_qubit + 1$

      $count\_qubit = count\_qubit + 2$

    **end for**

    $qubit\_init = qubit\_init + 1$

    $num\_gates = num\_gates - 1$

  **end for**

---

To enhance the accessibility and comprehensibility of the algorithm lecture, we have illustrated the circuit for the scenario where $n = 8$ in Fig.6. This visualization aims to make the algorithm more user-friendly and easier to use.

$$
\begin{array}{llll}
|x=0\rangle & \phantom{\times} & & |x=0\rangle \\
|x=1\rangle & \times & & |x=2\rangle \\
|x=2\rangle & \times & \times & |x=4\rangle \\
|x=3\rangle & \times & \times & \times & |x=6\rangle \\
|x=4\rangle & \times & \times & \times & |x=1\rangle \\
|x=5\rangle & \times & \times & & |x=3\rangle \\
|x=6\rangle & \times & & & |x=5\rangle \\
|x=7\rangle & & & & |x=7\rangle
\end{array}
$$

Figure 6: Qubit sorting circuit for the case of $n = 8$ qubits. Here, the fermionic SWAP gate has been represented using the same diagrammatic symbol as the SWAP gate.

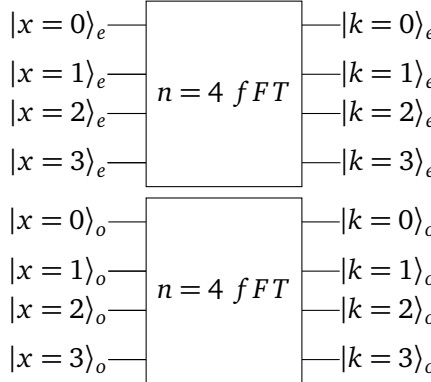

Figure 7: The $\frac{n}{2}$ fermionic Fourier transform circuit for the case of $n = 8$ qubits, the $e$ subindex stands for even while $o$ stands for odd. We use the periodicity of the Fourier transform, where the $k = 3$ state is equivalent to $k = -1$ state.

## $\frac{n}{2}$ fermionic Fourier transform

The next step involves applying two fermionic Fourier transforms to $\frac{n}{2}$ qubits, separately for the odd and even sectors. As a result, the transformed vector states correspond to momentum states labeled by $k$, ranging from $-\frac{n}{4} + 1$ to $\frac{n}{4}$ included.

It is important to highlight that Fourier space is periodic, specifically with a period of $\frac{n}{2}$. This periodicity allows the $k$ states to also be labeled from 0 to $\frac{n}{2}$. In the specific case where $\frac{n}{2} = 2$, the fermionic Fourier transform reduced to the application of $F_0^2$, as described by Eq.(60). Figure 7 illustrates the circuit scheme for the case of $n = 8$.

### Fourier states reorganization

The reorganization phase is designed to group the newly obtained $\frac{n}{2}$-qubit Fourier states by pairing together the $k$ states from the even sector with the corresponding $k$ states from the odd sector. This is achieved by the inverse circuit developed in the qubit sorting step. Figure 7 illustrates the resulting circuit for the $n = 8$ case.

### General Fourier transform gate application

At this stage, although we have obtained $k$ states resulting from the $\frac{n}{2}$-qubit fermionic Fourier transform, we still need to recover the $k$ states for the full $n$-qubit fermionic Fourier transform. To achieve this final step, the $F_k^n$ gate must be applied to the $|k_e\rangle$ and $|k_o\rangle$ states. This operation recovers the $|k\rangle$ and $|k + \frac{n}{2}\rangle$ states associated with the $n$-qubit Fourier transform.

$$
\begin{array}{ll}
|k=0\rangle_e & |k=0\rangle_e \\
|k=1\rangle_e & |k=0\rangle_o \\
|k=2\rangle_e & |k=1\rangle_e \\
|k=3\rangle_e & |k=1\rangle_e \\
|k=0\rangle_o & |k=2\rangle_e \\
|k=1\rangle_o & |k=2\rangle_o \\
|k=2\rangle_o & |k=3\rangle_e \\
|k=3\rangle_o & |k=3\rangle_o
\end{array}
$$

Figure 8: The Fourier states reorganization circuit for the case of $n = 8$ qubits, the $e$ subindex stands for even while $o$ stands for odd. Additionally, the SWAPs represented are fermionic SWAPs.

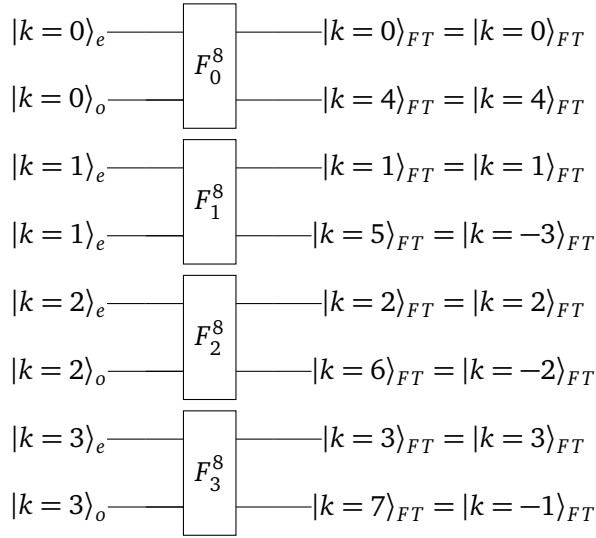

Figure 9: The General Fourier transform gate application circuit for the case of $n = 8$ qubits. Furthermore, we have illustrated the equivalence between the Fourier states, denoted by the $k$ labels ranging from $-\frac{n}{2}+1$ to $\frac{n}{2}$ or from $0$ to $n-1$.

---

**Algorithm 2** General Fourier transform gate application circuit

**Require:** $num\_qubit = 2^m$

**Ensure:** $qc\_generalFT \rightarrow$ Quantum circuit which recovers the $n$ Fourier transform states $|k\rangle$ and $|k+\frac{n}{2}\rangle$ from the $\frac{n}{2}$ Fourier transform states $|k_e\rangle$ and $|k_o\rangle$.

$\quad num\_qubit = 0$

$\quad$ **for** $k\_values = 0$ to $\frac{n}{2}-1$ **do**

$\quad\quad$ Add the $F_k^n$ gate to qubit $num\_qubit$ and $num\_qubit + 1$ with $k = k\_values$.

$\quad\quad num\_qubit = num\_qubit + 2$

$\quad$ **end for**

---

We have illustrated the circuit for the scenario where $n = 8$ in Fig.9, to clarify the algorithm described.

### 3.3   Bogoulibov transformation gate

The Bogoliubov transformation described in Eq.(47) mixes creation and annihilation operators from $k$ and $-k$ Fourier modes. Consequently, the vacuum changes after implementing the Bogoliubov transformation. The new vacuum state $|\Omega_0\rangle$ can be found in relation to the Fourier basis $|0\rangle, |k\rangle, |-k\rangle$ and $|-k, k\rangle$ imposing

$$
\begin{aligned}
a_k |\Omega_0\rangle &= \left( \cos\left(\frac{\theta_k}{2}\right)\alpha - \cos\left(\frac{\theta_k}{2}\right)\gamma b_{-k}^\dagger + i\sin\left(\frac{\theta_k}{2}\right)\delta b_{-k}^\dagger + i\sin\left(\frac{\theta_k}{2}\right)\alpha b_{-k}^\dagger b_k^\dagger \right)|0\rangle = 0\,, \\
a_{-k} |\Omega_0\rangle &= \left( \cos\left(\frac{\theta_k}{2}\right)\beta + \cos\left(\frac{\theta_k}{2}\right)\gamma b_k^\dagger - i\sin\left(\frac{\theta_k}{2}\right)\delta b_k^\dagger - i\sin\left(\frac{\theta_k}{2}\right)\beta b_k^\dagger b_{-k}^\dagger \right)|0\rangle = 0\,.
\end{aligned}
\tag{61}
$$

From the last equation, notice that $\alpha = 0$, $\beta = 0$ and $i\sin\left(\frac{\theta_k}{2}\right)\delta - \cos\left(\frac{\theta_k}{2}\right)\gamma = 0$. Using these constraints, the ground state is determined as

$$
|\Omega_0\rangle = \delta |0\rangle + \alpha |1_k\rangle + \beta |1_{-k}\rangle + \gamma |1_{-k} 1_k\rangle = \gamma \left( b_{-k}^\dagger b_k^\dagger + \frac{\cos\left(\frac{\theta_k}{2}\right)}{i\sin\left(\frac{\theta_k}{2}\right)} \right)|0\rangle\,,
$$

$$
\langle\Omega_0|\Omega_0\rangle = |\gamma|^2 \left( 1 + \frac{\cos^2\left(\frac{\theta_k}{2}\right)}{\sin^2\left(\frac{\theta_k}{2}\right)} \right)\,,
\tag{62}
$$

$$
|\gamma| = \sqrt{\frac{1}{1 + \frac{\cos^2\left(\frac{\theta_k}{2}\right)}{\sin^2\left(\frac{\theta_k}{2}\right)}}} = \sqrt{\sin^2\left(\frac{\theta_k}{2}\right)} = \left|\sin^2\left(\frac{\theta_k}{2}\right)\right|\,.
$$

Here, we have the freedom to choose the global phase of $\gamma$, we choose $\gamma = i\sin\left(\frac{\theta_k}{2}\right)$. Then the ground state is

$$
|\Omega_0\rangle = i\sin\left(\frac{\theta_k}{2}\right)|1_{-k} 1_k\rangle + \cos\left(\frac{\theta_k}{2}\right)|0\rangle\,.
\tag{63}
$$

Note that the new vacuum vector only depends on the vacuum of the FT and the $|1_k 1_{-k}\rangle$ Once the new vacuum is acquired, the remaining vectors can be derived by applying the creation operators $a_k^\dagger$ and $a_{-k}^\dagger$

$$
\begin{aligned}
a_k^\dagger |\Omega_0\rangle &= \left( \cos^2\left(\frac{\theta_k}{2}\right) b_k^\dagger |0\rangle + 0 \right) + \left( 0 + \sin^2\left(\frac{\theta_k}{2}\right) b_k^\dagger |0\rangle \right) = |1_k\rangle\,, \\
a_{-k}^\dagger |\Omega_0\rangle &= \left( \cos^2\left(\frac{\theta_k}{2}\right) b_{-k}^\dagger |0\rangle + 0 \right) + \left( 0 - \sin^2\left(\frac{\theta_k}{2}\right) b_k b_{-k}^\dagger b_k^\dagger |0\rangle \right) = |1_{-k}\rangle\,, \\
a_{-k}^\dagger a_k^\dagger |\Omega_0\rangle &= \cos\left(\frac{\theta_k}{2}\right)|1_{-k} 1_k\rangle + i\sin\left(\frac{\theta_k}{2}\right)|0\rangle\,.
\end{aligned}
\tag{64}
$$

Consider that the calculations have been done assuming the order $|-k, k\rangle$. However, for our purposes, it is more advantageous to rephrase this sequence as $|k, -k\rangle$, which entails incorporating a $-1$ whenever the state $|11\rangle$ is interchanged. The ultimate expressions are

$$
\begin{aligned}
|0_k 0_{-k}\rangle_{Bog} &= \cos\left(\frac{\theta_k}{2}\right)|0_k 0_{-k}\rangle_{FT} - i\sin\left(\frac{\theta_k}{2}\right)|1_k 1_{-k}\rangle_{FT}\,, \\
|1_k 0_{-k}\rangle_{Bog} &= |1_k 0_{-k}\rangle_{FT}\,, \\
|0_k 1_{-k}\rangle_{Bog} &= |0_k 1_{-k}\rangle_{FT}\,, \\
|1_k 1_{-k}\rangle_{Bog} &= -i\sin\left(\frac{\theta_k}{2}\right)|0_k 0_{-k}\rangle_{FT} + \cos\left(\frac{\theta_k}{2}\right)|1_k 1_{-k}\rangle_{FT}\,,
\end{aligned}
\tag{65}
$$

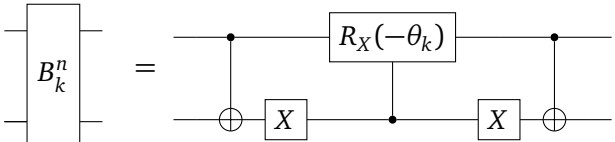

Figure 10: Decomposition of the building block of $B_k^n$ shown in Eq.(66), where $\theta_k$ is defined in Eq.(48).

where we have used a different notation. The $|k, -k\rangle_{Bog}$ corresponds to the Bogoulibov states, and the $|k, -k\rangle_{FT}$ corresponds to the Fourier states.

Once we have the Bogoulibov states written in terms of Fourier states, deriving the matrix that performs this operation is straightforward. Through the remainder of this work, we will refer to this matrix as the "Bogoulibov 2-qubit gate" or $B_k^n$. It takes the following form

$$
B_k^n = \begin{pmatrix} \cos\left(\frac{\theta_k}{2}\right) & 0 & 0 & i\sin\left(\frac{\theta_k}{2}\right) \\ 0 & 1 & 0 & 0 \\ 0 & 0 & 1 & 0 \\ i\sin\left(\frac{\theta_k}{2}\right) & 0 & 0 & \cos\left(\frac{\theta_k}{2}\right) \end{pmatrix},
\tag{66}
$$

where the $B_k^n$ matrix transforms the $|k, -k\rangle_{FT}$ vectors into $|k, -k\rangle_{Bog}$ and $\theta_k$ is described in Eq.(48).

The basic gate decomposition of $B_k^n$ is shown in Fig.10.

The circuit is designed to decouple the $k$ and $-k$ Fourier modes using the 2-qubit Bogoliubov gate, denoted as $B_k^n$. Although this task might initially seem straightforward, its complexity increases considerably when linear connectivity is taken into account.

This added complexity stems from the requirement of additional fermionic SWAP operations to rearrange the output states produced by the fermionic Fourier transform. Initially, these states are grouped as $k$ and $k + \frac{n}{2}$. However, for the Bogoliubov gates to work, the states must be reorganized into pairs of $k$ and $-k$ states.

Next, we will describe the algorithm used to build the Bogoulibov transformation circuit assuming linear connectivity and that the first qubit is numbered 0. This circuit is decomposed into two subcircuits:

1. **Bogoulibov qubit sorting:** The circuit consists of a series of fermionic SWAPS gates with the aim of grouping $k$ and $-k$ states.

2. **Bogoulibov gate application:** The circuit performs the Bogoulibov transformation applying the Bogoulibov gate into the modes $k$ and $-k$.

**Bogoulibov qubit sorting**

The initial step involves segregating qubits into $k$ and $-k$ modes. This can be optimally achieved by employing $\frac{n}{4} - 1$ cascades of fermionic SWAP gates, arranged according to a specific geometric pattern.

The first cascade begins at qubit 3, followed by the next cascade, which starts at the succeeding qubit after the first four gates of the previous cascade have been applied. This sequencing is crucial for optimizing the circuit's depth. If the second cascade is initiated before the completion of the fourth gate in the previous one, it would result in an incorrect sorting of states. While there are other, more straightforward geometries that can be programmed, such as applying cascades sequentially, they do increase the overall circuit depth.

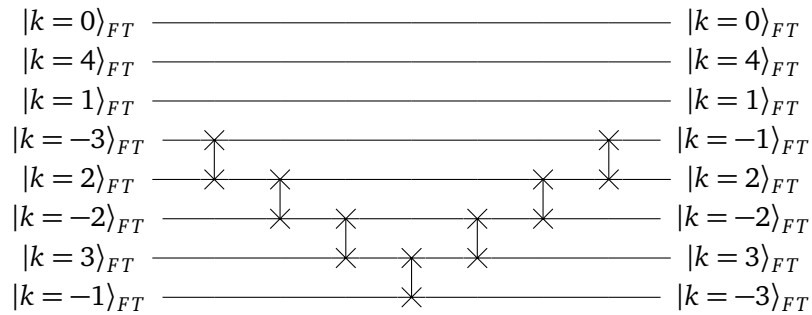

Figure 11: Bogoulibov qubit sorting circuit for the case of $n = 8$ qubits. Here, the fermionic SWAP gate has been represented using the same diagrammatic symbol as the SWAP gate.

Each cascade initially consists of $n - 4$ consecutive fermionic SWAP gates, each starting where the previous one left off. Subsequently, an additional $n - 5$ fermionic SWAP gates are applied sequentially, with each gate being applied one level above the previous one.

In Algorithm 3, we presented the algorithm in pseudocode:

---
**Algorithm 3** Qubit Sorting circuit

---
**Require:** $num\_qubit = 2^m$
**Ensure:** $qc\_Bog\_sorting \rightarrow$ Quantum circuit which groups the $k$ and $-k$ Fourier states.
  $num\_cascade = \frac{n}{4} - 1$
  $qubit\_init = 3$
  **if** $num\_cascade = 2$ **then**
    stop
  **else**
    **for** $i = num\_cascade$ to 1 **do**
      $count\_qubit = qubit\_init$
      $down\_cascade = i \cdot 4$
      $up\_cascade = (i \cdot 4) - 1$
      **for** $j = 1$ to $down\_cascade$ **do**
        Add fSWAP into $count\_qubit$ and $count\_qubit + 1$
        $count\_qubit = count\_qubit + 1$
      **end for**
      **for** $j = 1$ to $up\_cascade$ **do**
        Add fSWAP into $count\_qubit - 1$ and $count\_qubit$
        $count\_qubit = count\_qubit - 1$
      **end for**
      $qubit\_init = qubit\_init + 1$
      Start after the 4*th* fSWAP of the previous cascade
    **end for**
  **end if**

---

To enhance the accessibility and comprehensibility of the algorithm lecture, we have illustrated the circuit for the scenario where $n = 8$ in Fig.11. This visualization aims to make the algorithm more user-friendly and easier to use.

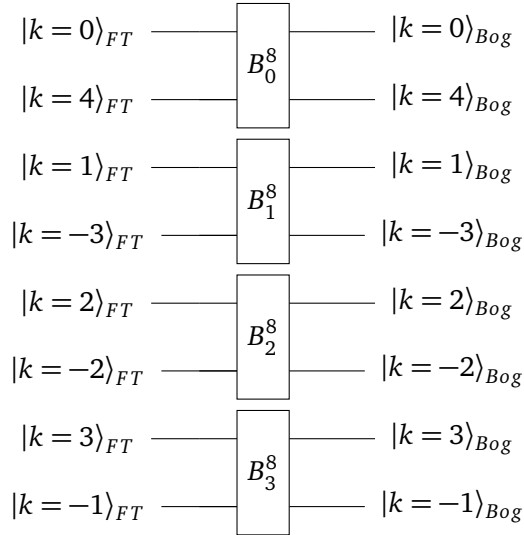

Figure 12: In the diagram is shown the Bogoulibov gate application circuit for the case of $n = 8$ qubits.

## Bogoulibov gate application

Finally, we have arrived at the last step to obtain our diagonalizing circuit. To disentangle the $k$ and $-k$ states, the Bogoulibov gate $B_k^n$ is applied. Hence, the new circuit will be simply a layer of Bogoulibov gates, where each gate will act on the corresponding $k$ and $-k$ states, starting from $k = 0$ to $k = \frac{n}{2} - 1$.

---

**Algorithm 4** Bogoulibov gate application circuit

---

**Require:** $num\_qubit = 2^m$
**Ensure:** $qc\_generalBog \rightarrow$ Quantum circuit which disentangles the $|k\rangle$ and $|-k\rangle$.
  $num\_qubit = 0$
  **for** $k\_values = 0$ to $\frac{n}{2} - 1$ **do**
    Add the $B_k^n$ gate to qubit $num\_qubit$ and $num\_qubit + 1$ with $k = k\_values$.
    $num\_qubit = num\_qubit + 2$
  **end for**

---

We have illustrated the circuit for the scenario where $n = 8$ in Fig.12, where can be stated the similarity with the general Fourier transform circuit.

## 3.4 Example: $n = 4$ and $n = 8$ spin chain

The explicit circuit $U_{dis}$ for spin chains with $n = 4$ and $n = 8$ is illustrated in Fig.13, Fig.14, and Fig.15. As an example of the many applications facilitated by $U_{dis}$, we performed simulations to evaluate the ground state and first excited state energies of the symmetric XY model ($J = 1$ and $\gamma = 0$) for spin chains of $n = 4$ and $n = 8$, considering various values of $\lambda$. Computing the energy of the ground and first excited state enables us to observe the quantum phase transition from an antiferromagnetic to a paramagnetic state, as mentioned in Sec.2. In the symmetric XY model, the diagonalized Hamiltonian becomes

$$\mathcal{H} = \sum_{k=\frac{-n}{2}+1}^{\frac{n}{2}} 2\left(\lambda + J\cos\left(\frac{2\pi k}{n}\right)\right) b_k^\dagger b_k - \lambda n, \tag{67}$$

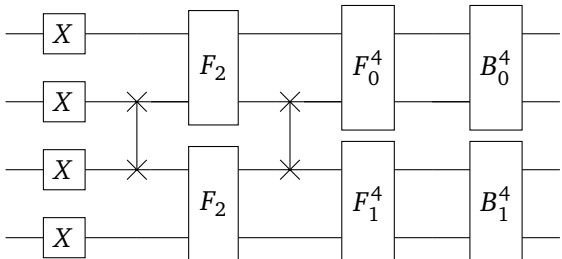

Figure 13: Quantum circuit $U_{dis}$ designed to diagonalize the XY Hamiltonian for $n = 4$ qubits. The initial layer consists of X gates, executing the Jordan-Wigner transformation. Subsequently, $F_2$ and $F_k^n$ implement the fermionic Fourier transform. The circuit concludes with the Bogoliubov transformation achieved by $B_k^n$. Additionally, the swaps represented in the diagram correspond to fermionic SWAPs.

where $b_k^\dagger b_k$ is the number occupation of the Fourier states $k$. Notice from Eq.(48), that in the case $\gamma = 0$ the Fourier and Bogoulibov modes are equivalents. For the case $n = 4$, the ground and the first excited state written in the diagonal basis are

$$|gs\rangle = \begin{cases} |0,1,0,0\rangle\,, & \lambda \leq 1\,, \\ |0,0,0,0\rangle\,, & \lambda \geq 1\,, \end{cases} \qquad |e\rangle = \begin{cases} |0,0,0,0\rangle\,, & \lambda \leq 1\,, \\ |0,1,0,0\rangle\,, & \lambda \geq 1\,. \end{cases} \tag{68}$$

For the case $n = 8$, the ground and the first excited state are

$$|gs\rangle = \begin{cases} |0,1,0,0,0,0,0,0\rangle\,, & \lambda \leq 1\,, \\ |0,0,0,0,0,0,0,0\rangle\,, & \lambda > 1\,, \end{cases} \qquad |e\rangle = \begin{cases} |0,0,0,0,0,0,0,0\rangle\,, & \lambda \leq 1\,, \\ |0,1,0,0,0,0,0,0\rangle\,, & \lambda > 1\,. \end{cases} \tag{69}$$

Additionally, we have simulated the ground state for the transverse field Ising model ($J = 1$ and $\gamma = 1$) in the $n = 4$ spin chain followed by the computation of the corresponding transverse magnetization. We have chosen magnetization because is one of the physical parameters which enable us to observe the phase transition discussed before. Analytically, the $\langle M_z \rangle = \sum_{i=1}^{n} \sigma_i^z$ in the ground state is

$$\langle gs|M_z|gs\rangle = \begin{cases} -\frac{\lambda}{2\sqrt{1+\lambda^2}}\,, & \lambda \leq 1\,, \\ -\frac{1}{2} - \frac{\lambda}{2\sqrt{1+\lambda^2}}\,, & \lambda \geq 1\,. \end{cases} \tag{70}$$

Notice that $M_z$ is not the conventional order parameter for the XY model [17, 18]. Nevertheless, we choose to compute its expectation value for several reasons. First, it enables direct comparison with previous studies on the topic [10, 11]. Second, although not the standard order parameter, $\langle M_z \rangle$ still captures the qualitative change in the ground state across the quantum phase transition. Finally, the Hamiltonian used in this work includes non-trivial boundary conditions which, while negligible in the thermodynamic limit, significantly affect the physics in finite-size systems. For small spin chains, such as those considered here, it is not evident that $M_x$ remains a valid order parameter, and a more detailed analysis would be required to justify its use in this context.

# 4 Time evolution

We have introduced the disentangling circuit $U_{dis}$ for the 1-D XY model, which enables us to obtain the complete spectrum of the Hamiltonian. This means that we can access the full physics of the system by applying the disentangling circuit to the computational basis. This

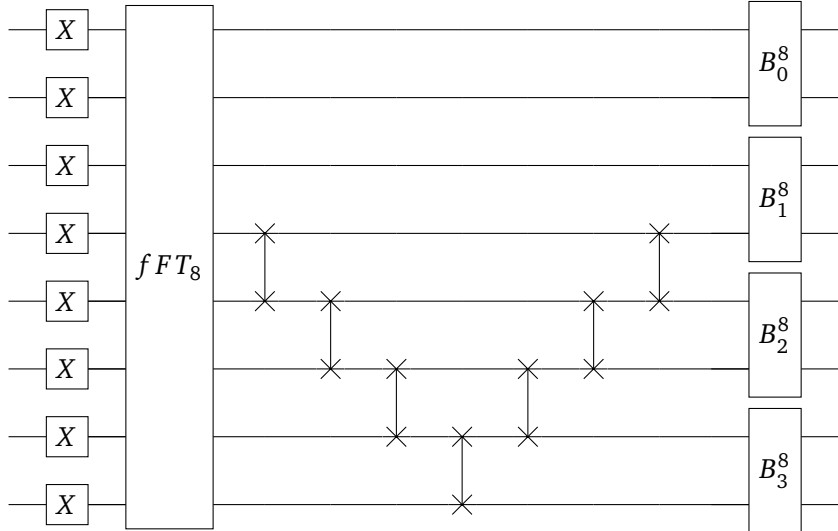

Figure 14: Quantum circuit $U_{dis}$ designed to diagonalize the XY Hamiltonian for $n = 8$ qubits. The initial layer consists of X gates, executing the Jordan-Wigner transformation. Subsequently, the fermionic Fourier transform is applied by the circuit $fFT_8$, described in Fig.15. The circuit concludes with the Bogoliubov transformation achieved by $B_k^n$. Additionally, the swaps represented in the diagram correspond to fermionic SWAPs.

approach also simplifies the calculation of various system properties, such as the expectation values of energy and magnetization, as discussed in the preceding section.

However, there are instances where our focus is on computing dynamic properties. In such cases, we need to calculate the time evolution of the state, which can be a challenging task. Nonetheless, a quantum circuit can be constructed to achieve exact time evolution for fermionic Hamiltonians that can be decomposed as the sum of the energies of each particle independently,

$$\mathcal{H} = \sum_{\alpha=1}^{N} \epsilon_\alpha a_\alpha^\dagger a_\alpha, \tag{71}$$

where $\epsilon_\alpha$ is the energy associated with having a particle in the state $\alpha$, $a_\alpha^\dagger$ and $a_\alpha$ are the fermionic creation and annihilation operator of the particle in the given state.

The reason for constructing the time evolution gate in this case is straightforward: for such Hamiltonians, the general time evolution operator $\mathcal{U}(t)$ can be decomposed into a product state of the time evolution operator for each qubit. To illustrate this, let's express the general time evolution of a given state $|\psi(t)\rangle$ driven by a non-time-dependent Hamiltonian $\mathcal{H}$. The evolution is accomplished by the unitary time-evolution operator $\mathcal{U}(t)$

$$\mathcal{U}(t) \equiv e^{-it\mathcal{H}},$$
$$|\psi(t)\rangle = \mathcal{U}(t)|\psi_0\rangle = \sum_l e^{-itE_l} |E_l\rangle \langle E_l| |\psi_0\rangle, \tag{72}$$

where $|\psi_0\rangle$ is the initial state, $|E_l\rangle$ are the eigenstates of the given Hamiltonian, and $E_l$ are the corresponding energies or eigenvalues of each state $|E_l\rangle$.

Due to the decomposable form of Hamiltonian in Eq.(71), eigenstates can be expressed as a product state of $N$ states, each representing the presence or absence of a fermion in the $\alpha$ state

$$|E_l\rangle = |\alpha = 1\rangle |\alpha = 2\rangle \cdots |\alpha = N\rangle, \tag{73}$$

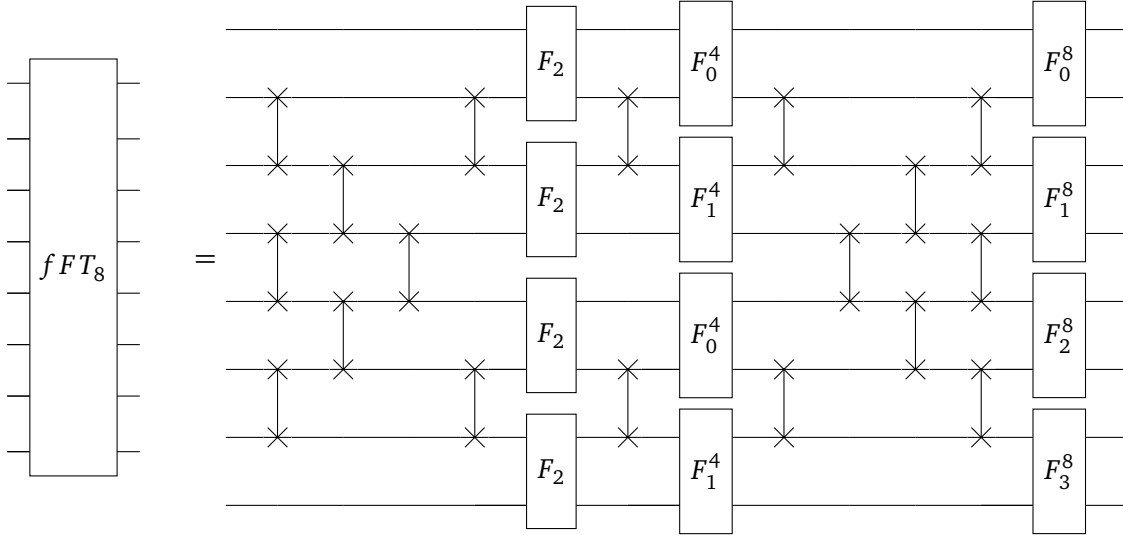

Figure 15: Fermionic Fourier transform circuit for the case $n = 8$. The swaps represented in the diagram correspond to fermionic SWAPs.

where $|\alpha\rangle$ can be represented by the qubits $|0\rangle$ or $|1\rangle$. Consequently, the time evolution operator becomes

$$
\begin{aligned}
\mathcal{U}(t)|E_l\rangle = e^{-it\mathcal{H}} |\alpha = 1\rangle |\alpha = 2\rangle \cdots |\alpha = N\rangle = e^{-it\sum_{\alpha=1}^{N} \epsilon_\alpha a_\alpha^\dagger a_\alpha} |\alpha = 1\rangle |\alpha = 2\rangle \cdots |\alpha = N\rangle \\
= \mathcal{U}_1 |\alpha = 1\rangle \mathcal{U}_2 |\alpha = 2\rangle \cdots \mathcal{U}_n |\alpha = N\rangle ,
\end{aligned}
\tag{74}
$$

where $\mathcal{U}_i$ is the time evolution operator for the $i_{th}$ qubit.

The procedure described above, tells us that to build the time evolution circuit we just need to perform time evolution for each qubit independently. Specifically for the XY 1-D model, the time evolution for the qubit representing a fermionic particle with momentum $k$ is

$$
\mathcal{U}_k = e^{-it2E_k a_k^\dagger a_k} e^{-it[-E_k + \epsilon_k - \lambda]} = U_1 U_2 ,
\tag{75}
$$

where $\epsilon_k = \lambda + J \cos(\frac{2\pi k}{n})$ and $E_k = \sqrt{\left(\lambda + J \cos\left(\frac{2\pi k}{n}\right)\right)^2 + \left(J\gamma \sin\left(\frac{2\pi k}{n}\right)\right)^2}$ are the energies associated to having one fermion in the Bogoulibov mode $k$.

The unitary operators $U_1$ and $U_2$ can be written in matrix form

$$
\begin{aligned}
U_1 &= e^{-it2E_k a_k^\dagger a_k} = \begin{pmatrix} 1 & 0 \\ 0 & e^{-it2E_k} \end{pmatrix} = \begin{pmatrix} 1 & 0 \\ 0 & e^{i\varphi_k} \end{pmatrix} , \\
U_2 &= \begin{pmatrix} e^{-it[-E_k + \epsilon_k - \lambda]} & 0 \\ 0 & e^{-it[-E_k + \epsilon_k - \lambda]} \end{pmatrix} = \begin{pmatrix} e^{i2\Phi_k} & 0 \\ 0 & e^{i2\Phi_k} \end{pmatrix} ,
\end{aligned}
\tag{76}
$$

whereby $E_k = \sqrt{\left(\lambda + J \cos\left(\frac{2\pi k}{n}\right)\right)^2 + \left(J\gamma \sin\left(\frac{2\pi k}{n}\right)\right)^2}$ and $\epsilon_k = \lambda + J \cos\left(\frac{2\pi k}{n}\right)$. Additionally, we have renamed the exponential arguments by $\varphi_k = -2tE_k$, and $\Phi_k = -2t[-E_k + \epsilon_k - \lambda]$. The gate decomposition of $\mathcal{U}_k$ is shown in Fig.16.

As an example of the many possibilities this gate opens, let's compute the time evolution of the expected value of transverse magnetization for the $n = 4$ qubits case, with $J = 1$ and $\gamma = 1$. Specifically, our initial state has all the spins aligned in the positive $z$ direction $|\psi(t = 0)\rangle = |\uparrow \uparrow \uparrow \uparrow\rangle$, which in the computational basis is written as $|0000\rangle$ state. The first step to compute the time evolution consists of expressing the initial state in the eigenbasis of

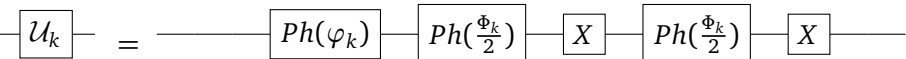

Figure 16: In the diagram is shown the decomposition of the building block of $\mathcal{U}_k$ shown in Eq.(76), where $\varphi_k = -2tE_k$, and $\Phi_k = -2t\left[-E_k + \epsilon_k - \lambda\right]$.

the XY Hamiltonian. This is achieved by precisely applying the $U_{dis}$ gate

$$|\psi(t=0)\rangle = \mathcal{U}_{dis}|0\ 0\ 0\ 0\rangle = \sin\left(\frac{\phi}{2}\right)|1\ 1\ 0\ 0\rangle - i\cos\left(\frac{\phi}{2}\right)|1\ 1\ 1\ 1\rangle, \qquad (77)$$

where $\phi = \mathrm{arctg}\left(\frac{1}{\lambda}\right)$. Subsequently, we apply the time evolution operator $U(t)$ to obtain $|\psi(t)\rangle$. Then, the time-dependent state is

$$
\begin{aligned}
|\psi(t)\rangle &= e^{-it2\left(\lambda-\sqrt{1+\lambda^2}\right)}\sin\left(\frac{\phi}{2}\right)|1\ 1\ 0\ 0\rangle - ie^{-it2\left(\lambda+\sqrt{1+\lambda^2}\right)}\cos\left(\frac{\phi}{2}\right)|1\ 1\ 1\ 1\rangle \\
&= e^{-it2\lambda}e^{-i2t\sqrt{1+\lambda^2}}\left(e^{+i4t\sqrt{1+\lambda^2}}\sin\left(\frac{\phi}{2}\right)|1\ 1\ 0\ 0\rangle - i\cos\left(\frac{\phi}{2}\right)|1\ 1\ 1\ 1\rangle\right),
\end{aligned}
\qquad (78)
$$

where the global phases are not physically relevant. After applying the time operator, we now apply the circuit $U^\dagger$ to obtain the state in the spin representation. Lastly, we compute analytically the expected value of the transverse magnetization $\langle M_z \rangle$, which yields the analytical result

$$\langle M_z \rangle = \frac{1 + 2\lambda^2 + \cos\left(4t\sqrt{1+\lambda^2}\right)}{2 + 2\lambda^2}. \qquad (79)$$

# 5 Results and discussion

In this section, we delve into the outcomes and insights derived from the application of our quantum circuit, $U_{dis}$, across various scenarios. The results show the classical simulation using the quantum computing library Qibo [9], for the spin chain $n = 4$ and $n = 8$ using the circuits represented in Figs. 13, 14, and 15.

Figure 17 presents the outcomes of the expected energy for the ground and first excited states in the symmetric XY model ($J = 1$. and $\gamma = 0$) for spin chains with $n = 4$ and $n = 8$. Given the nature of quantum simulations, subject to inherent probabilistic uncertainties, each data point carries a statistical error proportional to $\frac{1}{\sqrt{N}}$, where $N$ represents the number of shots—indicating the executions on a quantum processing unit (QPU). Here, $N$ was set to 1000. Notably, the results showcase the circuit's effectiveness in recovering analytical values for both cases. Moreover, a structural change in the ground state is evident at $\lambda = 1$, where the more stable state becomes the one without particles in the Bogoliubov modes $k$ instead of having a fermion in the $-k$ mode.

For the transverse field Ising model ($J = 1$ and $\gamma = 1$) in the $n = 4$ spin chain, the results of the ground state's expected value of transverse magnetization $\langle M_z \rangle$ are shown in Fig. 18. The circuit successfully reproduces analytical values, and at $\lambda = 1$, a magnetization discontinuity occurs due to a phase transition from an antiferromagnetic state to a paramagnetic state.

Moreover, we have also used the transverse field Ising model ($J = 1$ and $\gamma = 1$) to explore the time evolution of the expected value of transverse magnetization $\langle M_z(t) \rangle$. The quantum circuit $\mathcal{U}(t)$ is applied to evolve the initial state $|\uparrow, \uparrow, \uparrow, \uparrow\rangle$ with the magnetic field strength fixed at $\lambda = 0.5$. After, we apply $U_{dis}^\dagger$ to obtain the evolved spin state. The results are shown

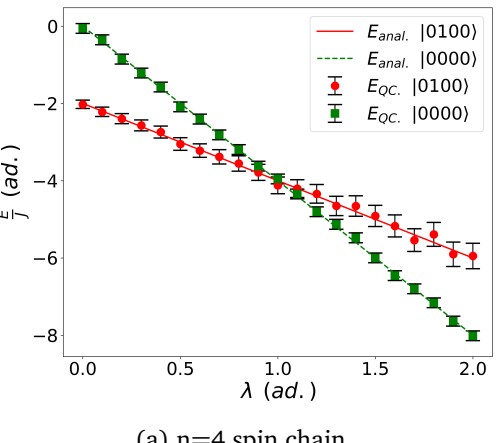

(a) n=4 spin chain.

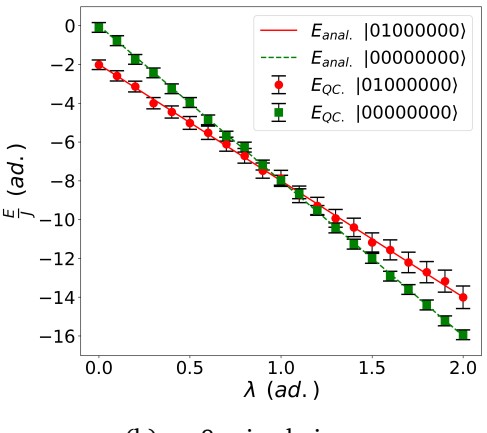

(b) n=8 spin chain.

Figure 17: Study of the ground and first excited state energy for the symmetric XY model ($J = 1$ and $\gamma = 0$) as a function of the transverse field strength parameter $\lambda$. The solid (dashed) line represents the analytical values of the energy $E$, while the scatter points correspond to results obtained from a quantum computer simulation conducted in Qibo. (a) shows results for an $n = 4$ spin chain, and (b) for an $n = 8$ spin chain.

in Fig.19, showcasing successful agreement between the quantum simulation and analytical values.

The circuit presented scales efficiently with the number of qubits. The Jordan-Wigner transformation is a simple layer of $X$ gate, as a result, escalates linearly with the number of qubits and the depth is constant. Similarly, the Bogoulibov transformation only combines $k$ and $-k$ modes, resulting in a constant circuit depth while the number of gates escalates proportionally to $\sim \frac{n}{2}$, where $n$ represents the number of qubits. In Ref. [27], it is shown that the circuit depth of the Fourier transforms follows a logarithmic scaling of $\sim \log_2(n)$, with the number of gates increasing as $\sim n \log_2(n)$. The time evolution circuit scales linearly with the number of qubits $n$ and presents a constant depth.

# 6 Conclusion

This paper presents a comprehensive implementation of the exact simulation of a 1-D XY spin chain using a digital quantum computer. Our approach encompasses the entire solution process for this exactly solvable model, involving key transformations such as the Jordan-Wigner transformation, fermionic Fourier transform, and Bogoliubov transformation. Additionally, we developed an algorithm to construct an efficient quantum circuit for powers of two qubits, capable of diagonalizing the XY Hamiltonian and executing its exact time evolution. The explicit code to reproduce these circuits is presented in Ref. [15] and uses Qibo, an open-source framework for quantum computing.

The presented quantum circuit is a powerful tool, facilitating the calculation of all eigenstate vectors by initializing qubits on a computational basis and subsequently applying the detailed circuit. This feature enables access to the complete spectrum of the Hamiltonian, providing novel approaches for exploring various system properties, including energy, magnetization, and time evolution.

Our introduced quantum circuit serves as a benchmark for quantum computing devices. It presents efficient growth and scalability with the number of qubits $n$, making it suitable to

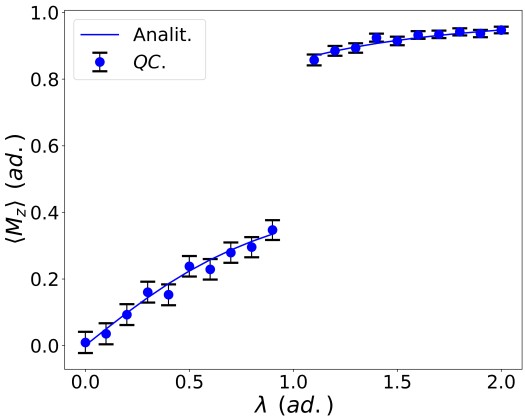

Figure 18: The ground state's expected value of transverse magnetization $\langle M_z \rangle$ for the transverse field Ising model ($J = 1$ and $\gamma = 1$) in a spin chain with $n = 4$ spins, as a function of the transverse field strength parameter $\lambda$. The solid line represents the analytical value of $\langle M_z \rangle$, while the scatter points correspond to the results obtained from a quantum computer simulation conducted in Qibo, utilizing the quantum circuit developed in this paper.

be used in devices of diverse sizes. Furthermore, the 1-D XY model's exact solvability not only allows us to test the efficiency of real quantum computers but it offers an avenue to study and model errors inherent in quantum computations, establishing a bridge between theoretical predictions and real-world outcomes.

Beyond its utility as a benchmark, the presented quantum circuit holds intriguing applications in condensed matter physics. The methods highlighted in this work can be extended to explore other integrable models, such as the Kitaev Honeycomb model [13], or with alternative Ansatz, as seen in the Heisenberg model [16].

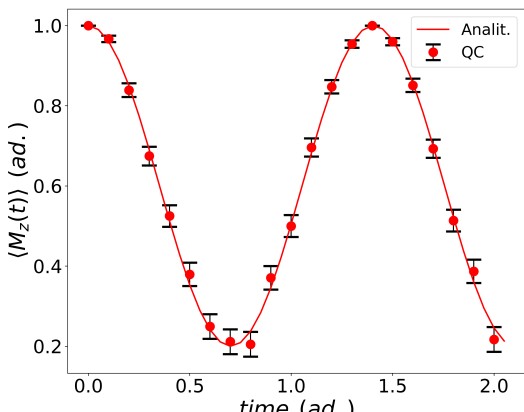

Figure 19: Time evolution simulation of transverse magnetization $\langle M_z \rangle$ for the transverse field Ising model ($J = 1$ and $\gamma = 1$) in a spin chain with $n = 4$ spins. The initial spin state is $|\uparrow, \uparrow, \uparrow, \uparrow\rangle$, evolved using the quantum circuit $\mathcal{U}(t)$ with the magnetic strength fixed at $\lambda = 0.5$. The solid line represents the analytical value of $\langle M_z \rangle$, while the scatter points correspond to the results obtained from a quantum computer simulation conducted in Qibo, utilizing the quantum circuit developed in this paper.

Moreover, different strategies for simulating thermal evolution [11] could be employed, paving the way for new approaches to studying quantum phase transitions. Notably, the XY Hamiltonian lacks an analytical solution in two dimensions, making it particularly interesting to use the circuit to simulate the 1D case as a foundation for constructing more sophisticated methods. For instance, this could serve as a stepping stone toward approximating the ground state of the 2D system. One potential avenue to achieve this would be introducing variational interactions within the circuit to capture the effects of the 2D Hamiltonian that are absent in the 1D case.

In conclusion, our work contributes to the advancement of quantum computing algorithms and establishes a foundation for exploring quantum solutions to complex problems in condensed matter physics.

# Acknowledgments

**Funding information**   A. C.-L. acknowledges funding from the Spanish Ministry for Digital Transformation and of Civil Service of the Spanish Government through the QUANTUM ENIA project call - Quantum Spain, EU through the Recovery, Transformation and Resilience Plan – NextGenerationEU within the framework of the Digital Spain 2026.

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
