# Peer review of "Simulation of the 1d XY model on a quantum computer"

_SciPost Physics Lecture Notes, doi:SciPost Phys. Lect. Notes 95 (2025)_

## Round 2 · Referee Report · Anonymous (Referee 1) · 2025-3-16

Report

The response from the authors looks good and I am OK with the revision.  I recommend publication.

Recommendation

Publish (easily meets expectations and criteria for this Journal; among top 50%)

---

## Round 2 · Referee Report · Anonymous (Referee 3) · 2025-3-20

Report

I appreciate the efforts made by the Authors in clarifying issues and remarks arisen during the previous round of referral.

I also agree with the decision of moving the submission to Lecture Notes, given the main "revisiting" character of the work.

I am still not fully happy with certain aspects of the manuscript, but they might in most cases be matter of taste and opinion. Indeed, other Referees seem to like the over-detailed derivation of equation in the style of an undergraduate lecture note. Therefore, I would not prevent publication based on my taste. This might even turn out as a good selling point for the manuscript.

I only add a few somehow minor points below for a final round of polishing.

Requested changes

1- Eq.(13) holds also trivially for i=j, so please remove "if i≠j"

2- Below Eq. (25) it might be worth to add that quadratic fermionic Hamiltonians are also naturally appearing in the mean-field treatment of more complicated systems (i.e., their realm of relevance is even wider than genuinely non-interacting models) :-)

3- Next sentence: fFT is only useful for translational invariant systems, whereas quadratic Hamiltonians are solvable exactly for whatever set of spatially dependent couplings (via Bogolubov Transformation and/or Fermionic Gaussian States). Excellent reviews are available, also on Scipost itself (and elsewhere, of course: Tagliacozzo and Santoro are just the first two senior authors that come to mind in this respect). Please address the reader correctly.

4- Sentence before Eq. (52): the H_XY Hamiltonian is already non-interacting! Maybe the Authors intended to say "convert into its diagonal form"?

5- P14L276: I would have thought that \ket{\pm} was a very common notation for (\ket{0} \pm \ket{1})/\sqrt{2}, what do the Authors mean here? I do not get the reasoning about \ket{0}... I am sure to overlook something easy that could be conveyed by rephrasing the sentence

6- Figure 18: The thing that was (and is still potentially) confusing is that the plot displays M_z, which is not the order parameter for the phase transition, and not M_x (or M_x^2), which instead is... but it is all fine, as long as the point is made clear in text. It escaped my attention in the first version, therefore I raised the issue back then. But how difficult and meaningful it is to plot also the more common order parameter itself?

Recommendation

Publish (meets expectations and criteria for this Journal)

  • validity: good
  • significance: ok
  • originality: ok
  • clarity: good
  • formatting: good
  • grammar: excellent

Author:  Marc Farreras  on 2025-05-16  [id 5487]

(in reply to Report 2 on 2025-03-20)
Category:
answer to question
reply to objection

We are thankful for the referee’s detailed report, and we consider that it has helped to improve the readability of the new manuscript version as well as clarify some of its parts.

Comments on the text:
“ Eq.(13) holds also trivially for i=j, so please remove "if i≠j"”
We sincerely appreciate the referee's suggestion. We acknowledge that distinguishing between the cases $i \eq j$ and $i \neq j$ is redundant. We have therefore removed this distinction in the revised version.

“Below Eq. (25) it might be worth to add that quadratic fermionic Hamiltonians are also naturally appearing in the mean-field treatment of more complicated systems (i.e., their realm of relevance is even wider than genuinely non-interacting models).”
“Next sentence: fFT is only useful for translational invariant systems, whereas quadratic Hamiltonians are solvable exactly for whatever set of spatially dependent couplings (via Bogolubov Transformation and/or Fermionic Gaussian States). Excellent reviews are available, also on Scipost itself (and elsewhere, of course: Tagliacozzo and Santoro are just the first two senior authors that come to mind in this respect). Please address the reader correctly.”
We thank the referee for pointing out this confusion in the text. We have addressed both issues in only one paragraph --> “Hamiltonians that are quadratic in fermionic creation and annihilation operators are ubiquitous in condensed matter physics. They describe systems of non-interacting fermions and also arise in the mean-field treatment of more complex interacting systems. Diagonalizing such Hamiltonians is a well-established procedure, typically accomplished using spatially dependent couplings and techniques such as the Bogoliubov transformation Ref.[Phys. Rev. B 53, 8486 (1996)] Ref.[J. Stat. Mech. 2011, P07015 (2011)] and fermionic Gaussian states Ref.[SciPost Phys. Lect. Notes 54 (2022),]. In translationally invariant models, such as the XY model, the Fourier transform is particularly useful, as it partially diagonalizes the Hamiltonian by making it local in momentum space. However, this transformation often introduces anomalous terms that couple different momentum modes, thus requiring a subsequent Bogoliubov transformation to achieve full diagonalization.”

“ Sentence before Eq. (52): the H_XY Hamiltonian is already non-interacting! Maybe the Authors intended to say "convert into its diagonal form"?”
We have corrected the mistake.

“P14L276: I would have thought that \ket{\pm} was a very common notation for (\ket{0} \pm \ket{1})/\sqrt{2}, what do the Authors mean here? I do not get the reasoning about \ket{0}... I am sure to overlook something easy that could be conveyed by rephrasing the sentence.”
We thank the referee for pointing out the potential confusion. We have revised the paragraph to make it clearer and less ambiguous--> “A second issue concerns a discrepancy in notation. In quantum computing, the spin states that are eigenstates of $sigma_z$ with positive and negative eigenvalues are conventionally denoted as $\ket{\uparrow} = \ket{0}$ and $\ket{\downarrow} = \ket{1}$, respectively. In contrast, in many-body physics, the symbol $\ket{0}$ (or sometimes $\ket{\Omega}$) typically denotes the vacuum state. Since the Jordan-Wigner maps $\ket{\downarrow}$ into $\ket{\Omega}$, an $X$ gate has been introduced to keep the standard convention and avoid potential confusion. As a result, the circuit is initialized with a layer of $X$ gates applied to each qubit. “

“Figure 18: The thing that was (and is still potentially) confusing is that the plot displays M_z, which is not the order parameter for the phase transition, and not M_x (or M_x^2), which instead is... but it is all fine, as long as the point is made clear in text. It escaped my attention in the first version, therefore I raised the issue back then. But how difficult and meaningful it is to plot also the more common order parameter itself?”
We thank the referee for pointing out the potential confusion regarding the choice of order parameter. Our initial focus was on analyzing and comparing with previous works, specifically Ref. [Phys. Rev. A 79, 032316 (2009)] and Ref. [Quantum 2, 114 (2018)], which primarily considers the expectation value of M_z. Following their approach, we did not initially examine the behavior of M_x or M_x^2 , which are the order parameters in the standard XY model.
Furthermore, the Hamiltonian we simulate includes boundary conditions that differ from the standard ones, and it only reduces to the conventional XY model in the thermodynamic limit (n->inf). For small spin chains, such as those we study, it is not trivial to assume that M_x remains a valid order parameter, and a more careful analysis would be required to justify this assumption.
Given that our simulations focus on small spin chains (e.g., n=4,n=8), our goal was not to characterize the order parameter in detail. Instead, we found it more illustrative to show that the circuit captures distinct ground state behavior as a function of the circuit parameters.
Finally, we would like to emphasize that our approach allows for the straightforward computation of various expectation values. For instance, M_x can be obtained as easily as M_z, simply by applying a Hadamard gate to each qubit before measurement to rotate the basis. Expectation values of M_x^2 can also be computed efficiently, and we indeed use such techniques to evaluate the ground state energy.
To clarify this in the text, we have added the following paragraph on page 25 before chapter 4 (after the analytical value of the transverse magnetization): Notice that $M_z$ is not the conventional order parameter for the XY model [Refs. 17, 18]. Nevertheless, we choose to compute its expectation value for several reasons. First, it enables direct comparison with previous studies on the topic Ref.[10,11]. Second, although not the standard order parameter, $\langle M_z \rangle$ still captures the qualitative change in the ground state across the quantum phase transition. Finally, the Hamiltonian used in this work includes non-trivial boundary conditions which, while negligible in the thermodynamic limit, significantly affect the physics in finite-size systems. For small spin chains, such as those considered here, it is not evident that $M_x$ remains a valid order parameter, and a more detailed analysis would be required to justify its use in this context.

---

## Round 2 · Referee Report · Anonymous (Referee 2) · 2025-5-6

Report

The authors have addressed my main concerns, and the paper is now much more readable. It still lacks any notable novelty, as it only relies on well-established techniques. Despite this, I still believe it could serve a pedagogical purpose, which makes it suitable for publication in SciPost Lecture Notes.

Recommendation

Publish (meets expectations and criteria for this Journal)

---

## Round 2 · List of Changes

Correction paper:

“In this paper, we present the comprehensive scheme”--> “In this paper, we present a comprehensive scheme”

In page 2 of the old version ( page 2 of the new version), the first paragraph, we have changed the last 5 lines “In this context, it has become ... to avoid the large accumulation of noise.” To contextualize better our work and include some references pointed by the referees.

“This paper presents a circuit specifically designed”-->”This paper presents a circuit suitable for the NISQ era”

“These transitions occur at absolute zero”--> “These transitions occur at zero temperature”
“spin leather operator”-->”spin ladder operator”

“The quadratic Hamiltonian in fermionic annihilation and creation operators appears in more condensed matter systems notably exemplified in the Hubbard model [17]. Diagonalizing this type of Hamiltonian is a well-established procedure, leading us to the subsequent phase: the fermionic Fourier transform (fFT).”-->”Hamiltonians quadratic in fermionic annihilation and creation operators are ubiquitous in condensed matter systems, describing systems of free fermionic particles. Diagonalizing this type of Hamiltonian is a well-established process, achieved through the fermionic Fourier transform (fFT).”

In page 12 of the old version (page 14 in the new one) we have changed the first paragraph “The second issue pertains to a notational problem. In conventional … different from the description provided in this work, although the final result should remain unchanged.” This change has been done with the aim of making the text clear and less confusing.

In page 13 of the old version ( page 15 of the new one), we have changed the first paragraph “At this point, we have understood the interplay … creation operators in the Fourier Transform definition.” This change has been done with the aim of making the text clear and less confusing.

In page 13 of the old version ( page 15 of the new one), we have changed the last paragraph “Up to this point, we have found … additional fermionic SWAPS become necessary.” This change has been done with the aim of making the text clear and less confusing.

In Figure 3. We have changed the footnote following the indications of the referee to improve the grammar and structure of the text.
In page 15 of the old version(page 18 of the new version), we have changed the paragraph in the subsection n/2 Fermionic Fourier Transform. The change aims to clarify some possible confusions that could arise in the old version.

In page 18 Of the old version (page 22 of the new version), we have changed the first paragraph “The circuit scheme involves the decoupling of k and -k Fourier modes … states needed for the Bogoulibov gates.” The change aims to clarify some possible confusions that could arise in the old version.

In page 23 of the old version (page 29 of the new version), we have added in the first paragraph of the section Results and discussion the phrase: “The results show the classical simulation using the quantum computing library Qibo [Ref], for the spin chain n=4 and n = 8 using the circuits represented in Figs.13, 14, and 15.” The aim of this change is to clarify that the results of the simulation are done by a classical simulator and not by a real quantum computer.

In the conclusions of the new version, we have added one paragraph in the end, expanding more about possible uses of this circuit, different than benchmarking, specifically how it can be used, for instance, as the building model of a variational ansatz to solve the 2D-XY model.

We have eliminated some intermediate steps in some of the equations to improve the clarity and readability of the text, as pointed out by the referees. The equations we have eliminated elements or changed are:
Eq.3; Eq.31-Eq.35;Eq.37;Eq.52; Eq. 61;Eq.64

---

## Round 3 · List of Changes

Under Eq.(25) we have added a more detailed explanation about the importance of quadratic Hamiltonians in physics and how to diagonalize them--> “Hamiltonians that are quadratic in fermionic creation and annihilation operators are ubiquitous in condensed matter physics. They describe systems of non-interacting fermions and also arise in the mean-field treatment of more complex interacting systems. Diagonalizing such Hamiltonians is a well-established procedure, typically accomplished using spatially dependent couplings and techniques such as the Bogoliubov transformation Ref.[Phys. Rev. B 53, 8486 (1996)] Ref.[J. Stat. Mech. 2011, P07015 (2011)] and fermionic Gaussian states Ref.[SciPost Phys. Lect. Notes 54 (2022),]. In translationally invariant models, such as the XY model, the Fourier transform is particularly useful, as it partially diagonalizes the Hamiltonian by making it local in momentum space. However, this transformation often introduces anomalous terms that couple different momentum modes, thus requiring a subsequent Bogoliubov transformation to achieve full diagonalization.”
Sentence before Eq. (52), we have exchanged the term “non-interacting form” by diagonal form.
We changed the first paragraph after Eq.(56) to make it more understandable -->”A second issue concerns a discrepancy in notation. In quantum computing, the spin states that are eigenstates of \(\sigma_z\) with positive and negative eigenvalues are conventionally denoted as \(\ket{\uparrow} = \ket{0}\) and \(\ket{\downarrow} = \ket{1}\), respectively. In contrast, in many-body physics, the symbol \(\ket{0}\) (or sometimes \(\ket{\Omega}\)) typically denotes the vacuum state. Since the Jordan-Wigner maps $\ket{\downarrow}$ into $\ket{\Omega}$, an $X$ gate has been introduced to keep the standard convention and avoid potential confusion. As a result, the circuit is initialized with a layer of $X$ gates applied to each qubit. “
In page 25, after Eq.(70) we have added a paragraph to explain the reasons why we simulate the transverse magnetization instead of the typical order parameter in the XY model-->”Notice that $M_z$ is not the conventional order parameter for the XY model [Refs. 17, 18]. Nevertheless, we choose to compute its expectation value for several reasons. First, it enables direct comparison with previous studies on the topic Ref.[10,11]. Second, although not the standard order parameter, $\langle M_z \rangle$ still captures the qualitative change in the ground state across the quantum phase transition. Finally, the Hamiltonian used in this work includes non-trivial boundary conditions which, while negligible in the thermodynamic limit, significantly affect the physics in finite-size systems. For small spin chains, such as those considered here, it is not evident that $M_x$ remains a valid order parameter, and a more detailed analysis would be required to justify its use in this context.”

---

## Editorial Decision

published